# JOINT OR DISJOINT: MIXING TRAINING REGIMES FOR EARLY-EXIT MODELS

## ABSTRACT

Early exits are an important efficiency mechanism integrated into deep neural networks enabling the network's forward pass to terminate early. These methods add trainable internal classifiers to the backbone network, which, however, changes the training dynamics. Most early exit methods either train the backbone network and exit heads simultaneously, or train the heads independently. However, the impact of this design choice on the overall network performance remains largely unexplored, as most studies simply select one approach without discussing its implications. In this paper, we analyze the effects of these training strategies on multi-exit networks, showing that joint training leads to impaired performance at higher computational budgets, while disjoint training results in suboptimal performance at lower budgets. To address these limitations, we propose a mixed training strategy where the backbone is trained first, followed by the training of the entire multi-exit network. Our results show that this alternative training regime arrives at solutions similar to standard static neural networks, yet does not share the disadvantages of disjoint training. We further analyze the differences between training regimes in terms of numerical rank, gradient dominance of each exit, and mutual information. Comprehensive evaluations across various architectures, datasets, and early-exit methods show consistent improvements in performance and efficiency using the proposed mixed strategy.

## 1 INTRODUCTION

Deep neural networks have achieved remarkable results across a variety of machine learning tasks. While the depth of these networks significantly contributes to their enhanced performance, the necessity of using large models for all inputs, especially in resource-constrained environments like mobile and edge computing devices, is questionable.

Early exit methods for deep neural networks have gained importance due to their potential to significantly improve computational efficiency. By exiting at earlier layers, these methods can decrease the number of operations needed for computation of the forward pass, leading to faster inference times. In doing so they allow the network to adapt its computational cost to the difficulty of the input sample. Simpler inputs can be processed with fewer layers, while more complex inputs can utilize the full capacity of the network.

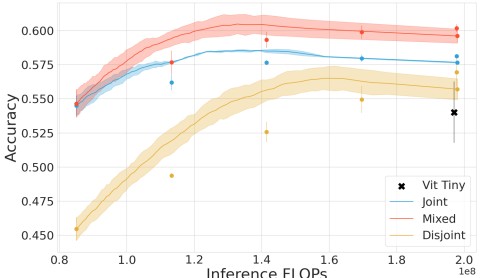

Figure 1: Performance-cost trade-off of the multi-exit network trained using three regimes considered in this paper. The choice of training regime impacts the performance across all computational budgets. (ViT / CIFAR-100).

Early exit methods are implemented through augmentation of the original architecture with internal classifiers (ICs) attached to selected intermediate layers (11). These ICs are designed to perform classification tasks based on the representations available at their respective positions in the network. A common approach for training early-exit models involves training the entire multi-exit network, including the added classifiers, from scratch (9; 33; 19) (**"joint" regime**). Alternatively, some methods train the backbone network first, then freeze its weights and train the parameters of the newly

added ICs in the second, separate phase of training (26; 15; 36) (**"disjoint" regime**). To the best of our knowledge, no study compares or explores the relationship between these training regimes.

In this study, we perform an extensive assessment of early-exit regimes and notice the choice of training strategy has a significant impact on the final model's performance, as can be seen in Fig. 1. We identify the relationship between computational budget and the choice of the regime. Using the disjoint regime results with a network that is significantly impaired when smaller computational budget is assumed. While the joint regime might initially seem as the appropriate way of training multi-exit networks, we demonstrate that due to its training dynamics it biases the network and produces a model with subpar performance on higher computational budgets.

In order to address these weaknesses of multi-exit networks, we propose a novel **"mixed" regime**: train the backbone network until convergence, then train the entire model jointly, including the internal classifiers, until convergence. This approach ensures that the backbone architecture is adequately trained before optimizing it alongside the internal classifiers for improved performance.

To gain a deeper understanding of learning and optimization in multi-exit architectures, we conduct an analysis of early-exit models trained under various regimes via mode connectivity, numerical rank and mutual information We also introduce the gradient dominance metric, and use it to reveal the set of ICs that have the largest impact on the backbone during training.

We provide a thorough empirical evaluation of early-exit regimes across different network architectures, data modalities, datasets and early-exit methods. Our results show that proposed alternative strategy enables significant improvements in performance in medium and high budgets over the commonly used joint training.

## 2  TRAINING REGIMES

Early exit methods fundamentally alter the organization of neural networks. It is widely believed that neural networks develop a hierarchical representation of features, where earlier layers learn basic shapes and patterns, while later layers progressively capture more complex abstractions (35). In other words, the earlier layers are characterized by higher frequency features while later layers learn low frequency elements. This regularity is disrupted in the case of early exit architectures as the backbone network is given additional classifiers that are placed in earlier parts of the network. These changes in architecture require a different approach for training and more nuanced analysis how the training should proceed.

In early-exit setting, the parameters can be divided into *backbone* parameters and *internal classifier (IC)* parameters. Each of these two groups of parameters can be trained separately, or jointly. In this paper, we frame the training process of any early-exit method as consisting of three following phases:

**Phase 1**: Train the backbone network parameters $\theta_b$ by minimizing the loss at the final output layer (could be the last IC or an added final classifier).

$$\theta_b^* = \arg\min_{\theta_b} \mathbb{E}_{(x_i, y_i) \sim \mathcal{D}} \left[ \mathcal{L}^{(K)}(\theta_b, \theta_{\mathrm{IC}}^{(K)}) \right] \tag{1}$$

During this phase, $\theta_{\mathrm{IC}}$ are either not present or not trained.

**Phase 2.** Train both the backbone network and the ICs simultaneously from scratch.

$$\theta^* = \arg\min_{\theta} \sum_{k=1}^{K} \alpha_k \mathbb{E}_{(x_i, y_i) \sim \mathcal{D}} \left[ \mathcal{L}^{(k)}(\theta_b, \theta_{\mathrm{IC}}^{(k)}) \right] \tag{2}$$

**Phase 3.** Freeze $\theta_b^*$ and train only the IC parameters $\theta_{\mathrm{IC}}$ .

$$\theta_{\mathrm{IC}}^* = \arg\min_{\theta_{\mathrm{IC}}} \sum_{k=1}^{K} \alpha_k \mathbb{E}_{(x_i, y_i) \sim \mathcal{D}} \left[ \mathcal{L}^{(k)}(\theta_b^*, \theta_{\mathrm{IC}}^{(k)}) \right] \tag{3}$$

In practical applications one can also use a set of pre-trained weights instead of training the model from scratch. Correspondingly, we generalize the early exit training regimes into three types based on which of the phases are performed:

**Disjoint training (Phase 1+3, "+-+").** The model parameters undergo training during the first and third phases, that is the backbone architecture is trained first, and then the ICs are trained separately with the backbone parameters being frozen.

**Joint Training (Phase 2, "-+-").** The training consists only of the second phase in which the entire model including the IC is trained from scratch. It is currently the most common way of training early-exit methods (18).

**Mixed training (Phase 1+2, "++-").** The training consists of two phases. The backbone is trained in isolation first, and then the entire network, including the ICs, is trained jointly. The regime emphasizes the importance of backbone pre-training as a better way to initialize the architecture for further training. This is our proposed way to improve early-exit training.

## 3 UNDERSTANDING MULTI-EXIT TRAINING REGIMES

In this section, we analyze the training dynamics of multi-exit networks. We examine the final models trained under the three training regimes, and investigate the gradient dominance of each head during training. In the appendix we also provide the loss landscape visualizations along with the details of the experimental setup.

### 3.1 MODE CONNECTIVITY

In this section, we demonstrate that the training dynamics of the proposed mixed regime differ from the commonly used joint regime, while the dynamics of the disjoint and mixed regimes are more similar.

We explore mode connectivity theory, which suggests that independently trained models often exhibit similar characteristics. Notably, after training two independent models, it is possible to find a continuous path in the parameter space where the loss remains low, enabling the models to be connected without encountering high-loss regions (6).

Building on the observation that independently trained neural networks can be linearly connected in weight space after accounting for permutation symmetries, as described in (1), we extend this idea to early-exit architectures trained under different regimes. Instead of focusing solely on independently trained networks, we investigate early-exit architectures trained in distinct regimes. Interestingly, the training regime plays a significant role in determining the mode connectivity between models.

While the model produced by the joint regime training occupies a different basin, the models trained in mixed and disjoint regime are much more closer to each other, as shown in Fig. 2. In fact, after accounting for permutation symmetries, the loss never gets high during linear interpolation of their weights, which indicates that they lie in the same basin.

This indicates that models trained under mixed and disjoint regimes find solutions that are not

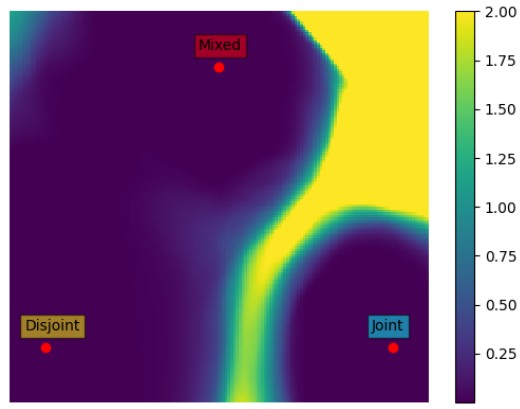

Figure 2: Mode connectivity between models trained with different training regimes . Colors represent the values of loss function, with yellow representing high loss ($\geq 2.0$). Disjoint and mixed regimes produce similar models, while the model trained in joint regime lies in a different basin (ResNet-20, CIFAR-10).

isolated. However, the disjoint regime is more constrained, as it trains only internal classifiers in the second phase. In contrast, the mixed regime benefits from the possibility of adapting the weights of the backbone to adjust for the added ICs, which leads to the overall lower loss of the model.

## 3.2 NUMERICAL RANK

While different regimes may fall into different loss basins, the question arises how that affects the learning outcome. To build a better understanding of multi-exit models, we look at how the choice of early-exit training regime influences intermediate representations of the backbone architecture. We analyze the expressiveness of early-exit architectures under different regimes by means of numerical ranks of activation maps (17). Mathematically, the rank is evaluated as:

$$r = \text{Rank}(A), \quad A \in \mathbb{R}^{n \times m} \tag{4}$$

where $A$ is the activation matrix of dimensions $n$ (number of samples) and $m$ (number of features).

The rank of the internal representations associated with different layers can provide insight into the "expressiveness", or capacity of the network. A higher rank (closer to the maximum possible for a given layer's matrix dimensions) indicates that the layer can capture more complex patterns or features in the data, as it implies a greater degree of linear independence among the feature detectors in that layer. High-rank activations matrices in a network suggest that the network is utilizing its capacity to learn diverse, high-frequency features, whereas a low rank might indicate that the network is not fully exploiting its potential.

In this framework, the numerical rank of the backbone architecture is analyzed under different early-exit regimes. A regular neural network is characterized by higher rank in earlier layers and lower rank in deeper layers as shown in Fig. 3a. Note that training only the backbone corresponds to the model obtained in the disjoint regime, as the backbone is not modified in this approach. In Fig. 3a we can see the change in network expressiveness after adding intermediate classifiers and training the entire architecture jointly. The numerical rank rises across the layers and becomes more uniform as consequence. This result indicates that optimal multi-exit models necessitates higher expressiveness of layers, and training with the disjoint regime prevents this from occurring.

Fig. 3b shows the difference between the network trained with mixed and joint regime. The mixed regime has a flatter structure with relatively lower ranks earlier and higher ranks later. We hypothesize this since early-exit architecture consists of a set of classifiers placed across the network, the flatter architecture is desired for better performance across all the classifiers. In contrast, a steeper curve as in the case of joint regime may resemble more a regular architecture and be less suitable for early-exit task.

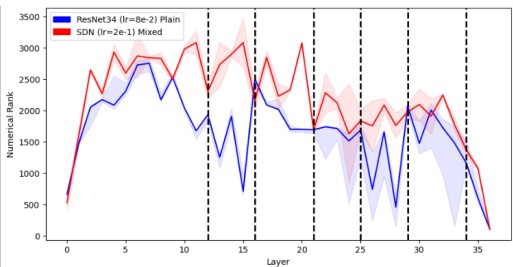 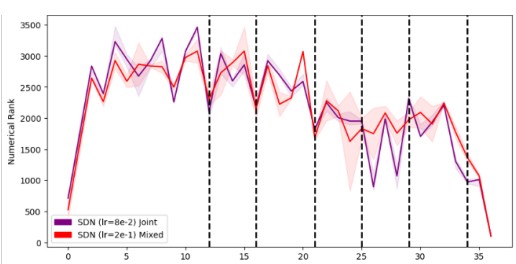

(a) The change in expressiveness of the network from Phase 1 (backbone) to Phase 2 (backbone+ICs).

(b) Mixed vs joint training. The vertical lines indicate IC placement.

Figure 3: Numerical ranks of backbone network in early-exit architecture trained with different regimes.

## 3.3 GRADIENT DOMINANCE

The use of internal classifiers during training in joint or mixed regime fundamentally alters the training dynamics, as these classifiers contribute to the overall loss. The gradient update now comes from multiple classifiers instead of just the final one, as in a standard neural network.

This leads to the following question: which gradients contribute the most to the overall gradient, and how do the gradients from different classifiers align? To answer this question, we first introduce a

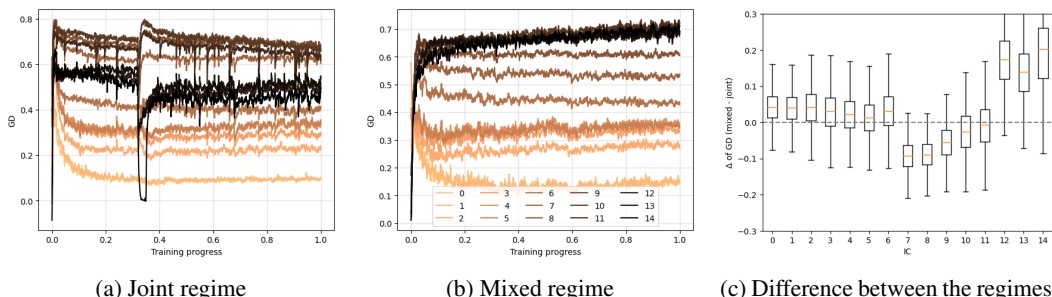

(a) Joint regime        (b) Mixed regime        (c) Difference between the regimes.

Figure 4: Gradient dominance for different regimes. Each line indicate how well gradients from different ICs align with the total gradient over course of the training. The last IC dominates the most in the mixed regime, which explains its excellent performance on higher computational budgets (ResNet-50, Tiny ImageNet).

metric called *gradient dominance*, which computes the cosine similarity between gradient from an individual internal classifier, $\mathbf{g}_{IC}$, and the overall gradient, $\mathbf{g}_{total}$:

$$\text{gradient dominance}(\mathbf{g}_{IC}, \mathbf{g}_{total}) = \frac{\mathbf{g}_{IC} \cdot \mathbf{g}_{total}}{\|\mathbf{g}_{IC}\|\|\mathbf{g}_{total}\|}$$

Gradient Dominance measures the consistency of the gradient directions produced by the early-exit classifiers and evaluates how well gradients from separate classifiers align with the overall gradient across the entire model. If the cosine similarity is close to 1, the auxiliary classifier's gradient is highly aligned with the total gradient, indicating that it potentially dominates other ICs with lower alignment in its impact on the total gradient.

In Fig. 4 we plot the Gradient Dominance for mixed and joint regimes, highlighting the difference in the training process between the two. In the mixed regime, deeper classifiers dominate the gradients, meaning that these early layers are more optimized to support the learning objectives of the later classifiers, potentially at the expense of earlier ones. Conversely, in the joint regime, the optimization tends to favor the subnetworks in the middle, where the gradients from the closest classifiers have a stronger impact.

Consequently, dominating gradients indicate that mixed regime is better optimized for samples that exit by deeper heads while joint for samples that are easier to classify at earlier ICs. Note that this observation is in line and with the study of information flow and explains the results of our empirical evaluation. The fact that the gradient of the last IC dominates the total gradient during almost the entire training period in the mixed regime explains why it never leaves the basin that models trained with the disjoint regime occupy.

### 3.4 MUTUAL INFORMATION.

In the context of neural networks, mutual information between $X$ and $Z$ represents how much information the input $X$ provides about the internal representation $Z$ after passing through a neural network. For random variables $X$ and $Z$, the mutual information is defined as: $I(X; Z) = \int_{x \in \mathcal{X}} \int_{z \in \mathcal{Z}} p(x, z) \log \frac{p(x,z)}{p(x)p(z)} \, dx \, dz$ where $p(x, z)$ is the joint probability distribution of $X$ and $Z$, and $p(x)$ and $p(z)$ are the marginal distributions of $X$ and $Z$, respectively. In practical terms, for neural networks, we use Monte Carlo sampling to estimate $I(X; Z)$ due to the high dimensionality of feature spaces.

In their work, (10) utilize the concept of mutual information between $X$ and $Z$ ($I(X; Z)$) in the framework of the information bottleneck (IB) principle. The IB principle aims to find a balance between the informativeness of the representation $Z$ for predicting the target variable $Y$ and the complexity of $Z$ in terms of its mutual information with the input $X$. Specifically, minimizing $I(X; Z)$ reduces the complexity and overfitting by ensuring $Z$ retains only the essential information from $X$, and maximizing $I(Y; Z)$ ensures that the representation $Z$ is informative enough to predict the target variable $Y$ effectively.

Early-exit architectures attach intermediate classifiers (ICs) to internal layers, altering the distribution of information flow as seen in Fig. 5. The effect is two-fold and differs between earlier and later layers. *Earlier layers.* The mutual information between $X$ and $Z$ is *larger* compared to a network trained without additional classifiers. *Deeper layers.* The mutual information for early-exit architecture is *lower* in the final layers.

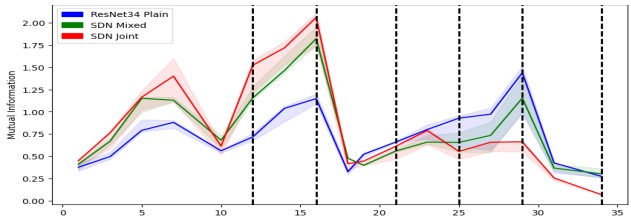

Figure 5: Mutual information $I(X; Z)$ between the input $X$ and the internal representation $Z$ of the backbone at different layers of the network for the three considered multi-exit model training regimes.

The above effect is seen in both regimes but is more pronounced in the joint regime. The information flow in the joint regime is more skewed and different from backbone-only training. Backbone training in the mixed regime makes the information flow fall between backbone-only and joint training. This is due to the fact that the representation of easy samples is not complex (that is, it is processed with just a few layers before exiting through an early IC). As the sample is easy, it is clearly and distinctly located within the boundaries of a single class. To describe it in terms of mutual information, the network does not need to reduce the complexity of $X$ to fit the internal representation $Z$, as $X$ has little irrelevant details. Consequently, the input $X$ is not compressed and the internal representation $Z$ has similar complexity to the representation of $X$, hence $I(X; Z)$ is higher.

Following this observation, we note that with higher $I(X; Z)$ in earlier layers, the joint strategy is more suitable for easy datasets where more samples exit at earlier layers. Similarly, the mixed regime learns more uniform representation of the $I(X; Z)$ across the network (one may observe an analogy to the numerical rank results in the Sec. 3.2) and may be preferred for more difficult datasets that exit at later internal classifiers.

## 4 EMPIRICAL EVALUATION OF TRAINING REGIMES

**Experimental set-up.** In this section, we perform tests on the commonly used simple early-exit method SDN (11), in which sample exits early if the confidence of a classifier is larger than a predefined threshold. We also include, MSDNet – a convolutional neural network architecture designed specifically for multi-exit models (9). In the next section we test regimes on a range of early-exit methods. To ensure proper model training, we test different learning rates for pre-training the backbone and separately for the next phase of training in each regime, and select the optimal one for each phase. For proper model convergence in each phase we utilize early-stopping to select a termination point of the training. This procedure halts training if there is no improvement in validation set accuracy over a specified number of epochs. In appendix we include all the experimental details for better reproducibility.

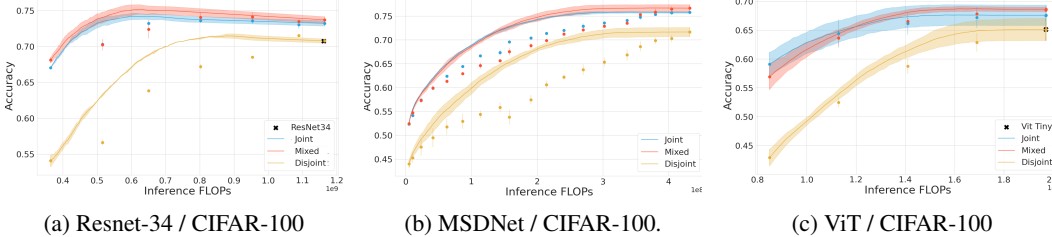

| (a) Resnet-34 / CIFAR-100 | (b) MSDNet / CIFAR-100. | (c) ViT / CIFAR-100 |

Figure 6: Comparison of training regimes for computer vision tasks. See more results in Appendix.

**Evaluation plots.** In all figures in this section, we assess a model by examining the trade-off between computational cost (FLOPs) and task performance (accuracy). The plots allow to see the performance of a training regime across a range of computational budgets.

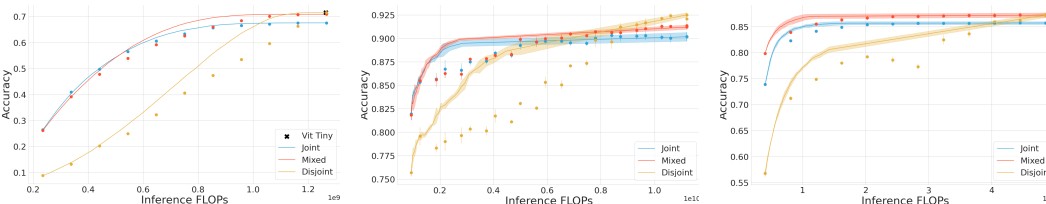

Figure 7: Comparison of training regimes for ViT-T / ImageNet-1k dataset.

Figure 8: Comparison of training regimes for BERT / SST-2 dataset.

Figure 9: Comparison of training regimes for BERT / Newsgroups dataset.

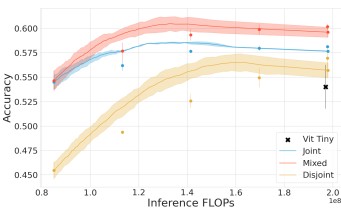
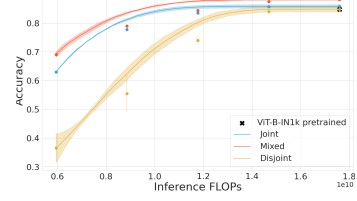
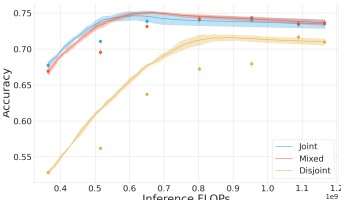

Figure 10: GPF (15) method implemented ViT and trained on CIFAR-100.

Figure 11: Performance of ViT-B pretrained on ImageNet-1k and fine-tuned on CIFAR-100.

Figure 12: Alternative exit criterion based on entropy. Training ResNet-34 on CIFAR-100.

To generate a plot, we set 100 evenly spaced early-exit confidence thresholds, and for each one we evaluate the model on the test set. We record when a sample achieves confidence threshold and exits, and then average FLOPs incurred and classification accuracy over all the samples. The trade-offs are aggregated for each method as a line on a two-dimensional plot, accompanied by its standard deviation. The standard deviation is calculated by conducting the experiment four times with different seeds. In the presented figures we also plot points, each one representing the score of a model with an exit strategy statically fixed to exit at a single particular IC. That is, we force every sample to exit at that given IC (which determines the FLOP count) instead of exiting by the threshold criterion.

## 4.1 Efficiency trade-offs in Vision and Natural Language Processing.

In Fig. 6 we present the performance of training regimes in two vision settings. We perform experiments on a ResNet-34, (8), MSDNet (9), and vision transformer (ViT) (2) architectures. We attach multiple ICs to each non-multi-exit model. In all setups, the disjoint regime performs significantly worse than the other two regimes, particularly under lower computational budgets. This highlights the dissonance between the features learned in the early layers of the backbone architecture and the early IC layers, which are trained separately, and do not perform well when merged together. On the other hand, the mixed and the joint regimes perform competitively with mixed slightly but consistently outperforming the joint regime. Mixed regime is particularly effective for higher computational budgets where samples are more complex and leave by later classifiers. However, for extremely low-budgets, joint regime may be preferable.

To test whether the results change when the scale is increased, we perform the same experiment for a vision transformer model trained on the ImageNet-1k dataset (22). From the results, which we present in Figure 7, we can see that the findings are mostly similar, with the difference being that the disjoint regime has similar performance to the mixed regime for the highest computational budgets.

In Figures 8 and 9 we present the performance of the BERT (5) multi-exit models on two natural language classification tasks. The significant difference is that the disjoint regime has better performance on the later classifiers and achieves the highest accuracy on SST dataset among all the regimes. SST is arguably the simplest dataset used in our experiments in terms of the complexity of input data and the number of classes. Nevertheless, in both tasks the mixed regime presents improved results over the joint regime across most of the computational budgets.

## 4.2 VARYING EARLY-EXIT METHODS, EXIT CRITERIA AND PRETRAINED MODELS

In Fig. 26 we present an example of an alternative method, GPF (15). In appendix we also include the experiments on PBEE (36) and ZTW (29). The results are consistent with the previous ones, and again highlight the improvement achieved by training the network in the proposed regime.

In Fig. 27 we repeat the experiment, but on the practical setup where we have a model pre-trained on a different dataset. Mixed regime still outperforms the remaining two regimes, and achieves superior performance even on the lowest computational budgets. This means that the mixed regime adapts well to a pre-training model and can can transfer the features in the course of the second phase joint training. This finding has positive benefits. It demonstrates the effectiveness of using a pre-trained model and facilitates the training of an early-exit architecture by enabling the use of a model trained on a different dataset, even if the dataset is not directly accessible.

Finally, we compare different exit strategies. In all experiments, we employ the commonly used maximum softmax confidence criterion, which triggers an exit when the probability of the most likely class exceeds a certain threshold. In Fig. 22, we also present results for an alternative strategy based on the entropy exit criterion. Entropy is computed over the predicted probability distribution from the neural network's softmax output, and an exit occurs when the entropy surpasses a specified threshold. Though the mixed regime exhibits a slight decline for lower computational budgets, the conclusions from the other experiments still hold.

## 4.3 PROPER BACKBONE TRAINING

In this work, we argue that training the backbone first has a beneficial effect on the early-exit architecture performance. In this section, we look at the effects of undertraining the backbone in the first phase of training on the performance-score trade-off results of models trained with the mixed regime.

As shown in Fig. 13, undertraining the backbone negatively affects mixed training setting in a significant way. We perform the same experiments as previously for our ViT-B model, but with the backbone trained with a lower, unoptimal learning rate, which results in an undertrained backbone after the first phase that achieves on average 77% accuracy. In such case mixed training underperforms and joint training yields a better outcome. This highlights the importance of training the backbone properly.

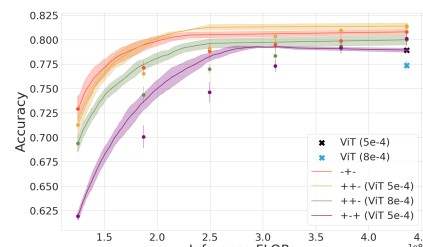

Figure 13: Performance of models trained in the mixed regime drops with undertrained backbone (SDN, ViT, CIFAR-10).

## 4.4 IC DENSITY PLACEMENT

The density of placing the internal classifiers in early exit architectures refers to how often these classifiers are inserted at different layers within the neural network. This can range from being placed at every layer to being placed at strategic intervals, depending on the architecture and the specific use case.

This density influences the network's performance across different training regimes, as shown in Table 1. When ICs are placed after each layer, the mixed regime outperforms the joint regime, particularly when accounting for input variation. The mixed regime excels with frequent classifier placement, making it well-suited for inputs with varying complexity. As placements become less frequent, the difference between joint and mixed regimes becomes less pronounced. Nevertheless, the mixed regime generally remains superior, and the disjoint regime performs the worst overall. In appendix we perform a similar analysis for different head sizes.

## 4.5 IMPACT OF LOSS AND GRADIENT SCALING

Section 3.3 highlights how the mixed regime effectively emphasizes deeper intermediate classifiers. A comparable effect can be achieved by assigning larger coefficients to the losses of deeper ICs and

Table 1: The effect of varying head placement frequencies on the SDN early-exit architecture with ViT as a backbone, trained on Imagenette. Heads are placed after every $n$ layers. Given accuracy is obtained as in (29) using the time budget: 25%, 50%, 75%, 100% of the base network and without any limit (Max).

| $n$ | Regime | 25% | 50% | 75% | 100% | Max |
|---|---|---|---|---|---|---|
| 1 | mixed | 82.21 ±0.08 | 83.20 ±0.39 | 83.19 ±0.41 | 83.19 ±0.41 | 83.18 ±0.45 |
| | joint | 81.44 ±1.54 | 82.51 ±1.40 | 82.46 ±1.41 | 82.46 ±1.41 | 82.45 ±1.40 |
| | disjoint | 74.69 ±1.08 | 78.17 ±1.49 | 78.08 ±1.35 | 78.07 ±1.31 | 78.07 ±1.31 |
| 2 | mixed | 79.17 ±0.83 | 80.98 ±1.35 | 80.81 ±1.17 | 80.81 ±1.17 | 80.80 ±1.17 |
| | joint | 76.92 ±1.57 | 80.08 ±1.92 | 80.24 ±2.07 | 80.21 ±2.03 | 80.22 ±2.05 |
| | disjoint | 73.61 ±0.46 | 78.24 ±1.14 | 78.09 ±1.34 | 78.07 ±1.31 | 78.07 ±1.31 |
| 3 | mixed | 78.87 ±1.10 | 80.69 ±1.08 | 80.64 ±0.96 | 80.64 ±0.96 | 80.64 ±0.96 |
| | joint | 77.83 ±0.27 | 80.14 ±0.63 | 80.11 ±0.51 | 80.10 ±0.51 | 80.10 ±0.51 |
| | disjoint | 72.14 ±1.05 | 77.88 ±1.38 | 78.11 ±1.41 | 78.07 ±1.31 | 78.07 ±1.31 |

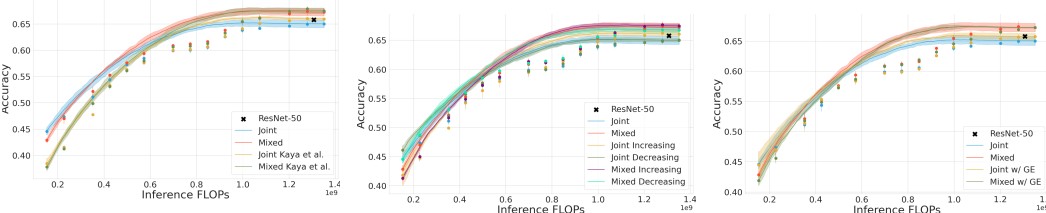

Figure 14: Multi-exit models trained with loss weights that change during training (11).

Figure 15: Multi-exit models with simple loss scaling (7).

Figure 16: Multi-exit models with gradient scaling (14).

smaller coefficients to earlier ones, as demonstrated in prior work (11; 7). Similarly, the gradient equilibrium method proposed by Li et al. (14) achieves this prioritization by attenuating the gradient magnitudes of earlier ICs. In this section, we revisit these approaches within both the joint and mixed regimes to further substantiate the advantages of the proposed mixed regime.

Kaya et al. (11) introduced a technique that linearly increases loss coefficients during training. In Figure 14, we evaluate this loss scaling method on a multi-exit ResNet-50 model trained on the TinyImagenet dataset. Our findings indicate that: (1) for the joint regime, loss scaling enhances performance at higher computational budgets but leads to reduced performance at lower budgets; and (2) **for the mixed regime, there is no observable improvement**, while performance at lower budgets is still reduced.

We extend our analysis to the constant loss weighting scheme proposed by Han et al. (7). Specifically, this approach maintains an average coefficient of 1, and the coefficients either increase or decrease linearly along model depth, with first and last coefficient being equal to 0.6 and 1.4. The results, shown in Figure 15, align with those observed in the previous experiment.

Finally, in Figure 16, we conduct a similar experiment using the gradient equilibrium method proposed by Li et al. (14). Consistent with the findings of the original study, gradient scaling enhances performance at higher computational budgets for the joint regime without compromising performance at lower budgets. However, for the mixed regime, there is again no observable improvement, highlighting that **the straightforward mixed regime training obviates the need for scaling techniques**.

## 5 RELATED WORK

Early exiting is a notable application of the conditional computation paradigm (3). While conceptually similar to earlier classifier cascades (32; 27), it differs in that all classifiers are integrated

within a single model, enabling end-to-end training. The first multi-exit model was introduced by Teerapittayanon et al. (26), and the field has expanded considerably since its inception.

Joint training is the most widely used and well-established strategy for early-exit models (18). This approach has been successfully applied to dynamic inference under various constraints, such as energy or time limitations (28), and extended to diverse early-exit applications, including low-resolution classification (31), quality enhancement (33), and Question-Answering systems (25). While joint training has proven effective, several studies have demonstrated significant improvements through modifications to the training process. For instance, knowledge distillation from the final classifier to earlier internal classifiers has been shown to enhance their performance (20; 14; 16). Similarly, ensembling multiple intermediate classifiers can improve prediction accuracy (21; 23). The Global Past-Future (GPF) method (15) incorporates information from both earlier predictions and surrogate later predictions to improve inference. Additionally, recent works (7; 34; 4) identify a train-test mismatch in conventional multi-exit approaches and propose strategies to address this issue, further enhancing the robustness of early-exit models.

SDN (11) was one of the first to explore the training of early-exit models through the pre-training of the architecture's backbone followed by separate training of the classifiers. Multiple subsequent works have focused on optimizing early-exit models based only on this setup (29; 12; 16), potentially limiting the general applicability of their findings. For instance, Wołczyk et al. (30) employ an ensembling technique that combines predictions from earlier internal classifiers, weights of which are trained in a separate, third training phase. Lahiany et al. (12) propose PTEENet, which augments pre-trained networks with confidence heads that dynamically adjust based on available resources and unlabeled data.

Kaya et al. (11) were the first to explore both joint and disjoint training approaches for early-exit models. These approaches are also briefly reviewed in surveys such as (24; 18). Furthermore, techniques like weighting the losses at each exit head (36; 11; 7) or scaling gradients (14) can be regarded as variations of the joint training paradigm. To the best of our knowledge, this work is the first to directly compare models trained under different regimes and to provide a detailed analysis of the training dynamics of multi-exit models.

## 6 CONCLUSION, DISCUSSION AND TAKEAWAYS

This study contributes insights into the training of early-exit models, providing a foundation for developing more efficient dynamic deep learning systems. The work presents a comprehensive analysis and evaluation of different training regimes for early-exit models in deep neural networks. By categorizing training approaches into disjoint, joint, and mixed regimes, we have demonstrated that the way the backbone and internal classifiers in early-exit architectures are trained influences its performance and efficiency. Below we summarize some practical takeaways when training early-exit architectures.

*Mixed*. Mixed regime demonstrates substantial robustness across various factors, including different data modalities and early-exit approaches with varying exit criteria. Therefore, the mixed regime is generally preferred, combining the benefits of both disjoint and joint training. The mixed regime ensures that the backbone network is well-optimized before integrating internal classifiers, leading to improved computational efficiency and accuracy. It is particularly recommended for cases where the performance on medium and higher computational budgets is the most important requirement.

*Joint*. Joint regime may be preferable as it is relatively simple to implement, and performs well for small computational budgets. However, it underperforms when the backbone is initialized from a pre-trained model.

*Disjoint*. This regime is generally inferior compared to the others in multiple setups, but performs well for some language datasets with a low number of classes. It may be preferred when the backbone is shared, or the lack of resources prevents us from training the backbone network.

Future research should investigate more sophisticated optimization techniques, such as adaptive learning rates or meta-learning strategies, specifically designed for early-exit models. Moreover, other training strategies may be proposed where training is tailored to particular sub-networks within the entire early-exit architecture.

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

# A    LOSS LANDSCAPE

The concept of a loss landscape in the context of neural networks is crucial for understanding the training dynamics and generalization properties of models. The loss landscape provides a visual and analytical representation of how the loss function changes with respect to the model's parameters. By visualizing the loss landscapes of different neural network architectures, we can understand how design choices affect the shape of the loss function.

For a trained model with parameters $\theta^*$, one can evaluate the loss function for the numbers $x, y$

$$f(x, y) = L(\theta^* + x\delta + y\eta) \tag{5}$$

such that $\delta, \eta$ are random directions sampled from a probability distribution, usually a Gaussian distribution, filter-normalized (13), obtaining a 3D plot. In contrast to a typical neural network architecture, in early-exit set-up, both the final and internal classifiers are considered. We consider total training loss and separate losses for each IC. When evaluating head losses, we use common random directions$(\delta, \eta)$ for each IC. The $\delta, \eta$ directions contain both backbone and head parameters.

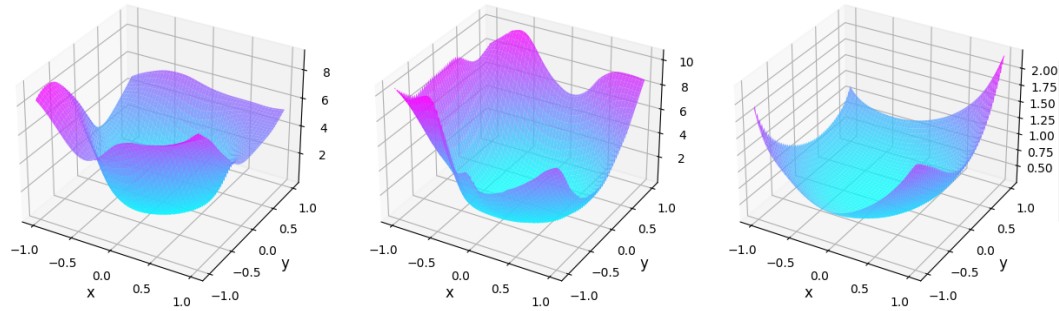

Figure 17: Training loss landscapes: comparison for Joint, Mixed, and Disjoint regimes (left to right), head 1. Landscapes for SDN architecture with Resnet20 as backbone, on CIFAR-10 dataset

As shown in Fig. 17 there is a significant difference in loss landscapes between the Disjoint regime and the Joint one. The Joint and Mixed regimes are similar in this regard.

In Fig. 13 we also include depictions for losses where $\delta$ and $\eta$ are sampled from uniform distribution. In this setting, mixed regime is characterized by smoother losses compared to joint regime showing that backbone pre-training may lead to easier optimization problem for early-exit architecture.

# B    IC SIZE

The size of an internal classifier in early exit architectures refers to the number of layers and neurons that make up the classifier inserted at intermediate layers of a neural network. The internal classifier typically consists of a linear layer, such as a fully connected (dense) layer or a small convolutional block, although there is no standard IC architecture in literature.

The size of the internal classifier directly affects the computational cost of the early exit. Smaller internal classifiers are computationally cheaper and faster, enabling quick early exits without significant overhead. Larger internal classifiers, while potentially more accurate due to their increased capacity, may negate some of the computational savings achieved by early exits, especially if they are nearly as large as the remaining layers of the network.

We examine the effect of varying head sizes for the SDN architecture with ViT as the backbone. Each head architecture consists of either one or two connected layers with output dimensions of 1024 or 2048, followed by a softmax layer. As shown in Table 2, smaller architectures outperform larger ones. However, for larger architectures, there is a decrease in variation when using the joint training regime.

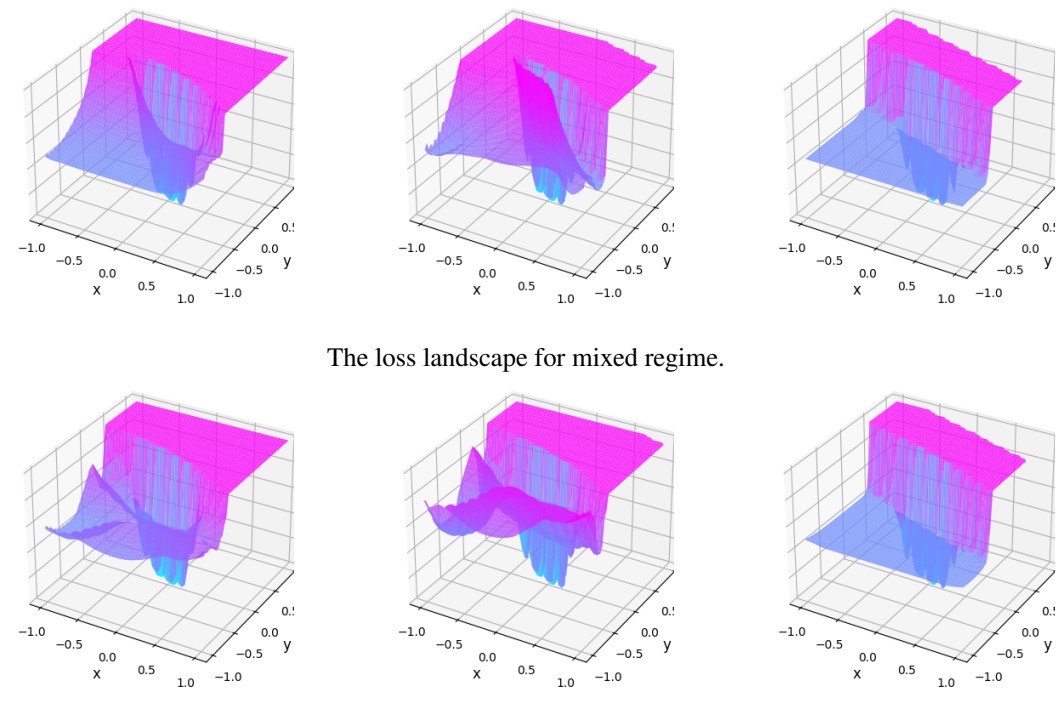

The loss landscape for mixed regime.

The loss landscape for joint regime.

Figure 18: The loss landscape with uniform sampling.

Table 2: The effect of varying head architecture size on the SDN early-exit architecture with ViT as a backbone, trained on Imagenette. Size describes number of layers used and output dimension.

| Size | Regime | 25% | 50% | 75% | 100% | Max |
|---|---|---|---|---|---|---|
| 1L-1024 | mixed | 82.21 ±0.08 | 83.20 ±0.39 | 83.19 ±0.41 | 83.19 ±0.41 | 83.18 ±0.45 |
| | joint | 81.44 ±1.54 | 82.51 ±1.40 | 82.46 ±1.41 | 82.46 ±1.41 | 82.45 ±1.40 |
| | disjoint | 75.61 ±0.10 | 78.72 ±1.24 | 78.32 ±1.62 | 78.29 ±1.63 | 78.28 ±1.65 |
| 2L-1024 | mixed | 80.25 ±0.59 | 81.28 ±0.44 | 81.13 ±0.40 | 81.13 ±0.40 | 81.11 ±0.37 |
| | joint | 80.00 ±0.69 | 81.06 ±0.57 | 80.99 ±0.69 | 80.99 ±0.69 | 81.00 ±0.68 |
| | disjoint | 77.41 ±0.39 | 79.04 ±1.26 | 78.36 ±1.63 | 78.29 ±1.63 | 78.28 ±1.65 |
| 2L-2048 | mixed | 79.63 ±0.67 | 80.77 ±0.58 | 80.70 ±0.51 | 80.70 ±0.51 | 80.67 ±0.52 |
| | joint | 79.68 ±0.28 | 80.67 ±0.08 | 80.64 ±0.15 | 80.64 ±0.15 | 80.61 ±0.13 |
| | disjoint | 77.10 ±0.40 | 79.02 ±1.26 | 78.39 ±1.69 | 78.30 ±1.65 | 78.28 ±1.65 |

## C  EARLY-EXIT METHODS

We include experiments for additional early-exit method, PABEE (36), which has also alternative exit-policy, and ZTW (29), which reuses predictions returned by its predecessors.

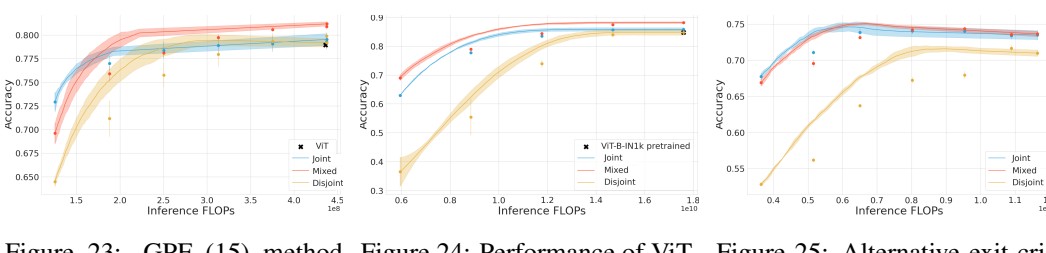

Figure 23: GPF (15) method implemented ViT and trained on CIFAR-10.

Figure 24: Performance of ViT-B pretrained on ImageNet-1k and fine-tuned on CIFAR-100.

Figure 25: Alternative exit criterion based on entropy. Training ResNet-34 on CIFAR-100.

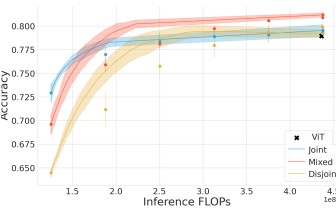
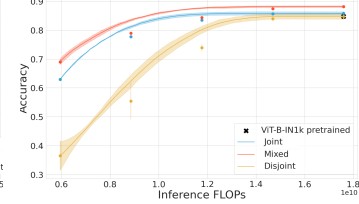
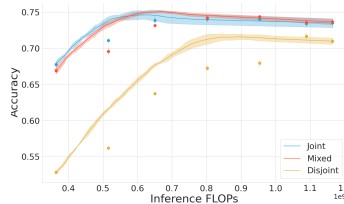

Figure 26: GPF (15) method implemented ViT and trained on CIFAR-10.

Figure 27: Performance of ViT-B pretrained on ImageNet-1k and fine-tuned on CIFAR-100.

Figure 28: Alternative exit criterion based on entropy. Training ResNet-34 on CIFAR-100.

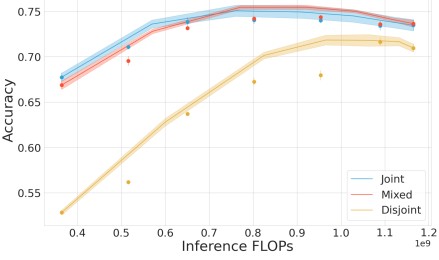
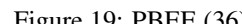
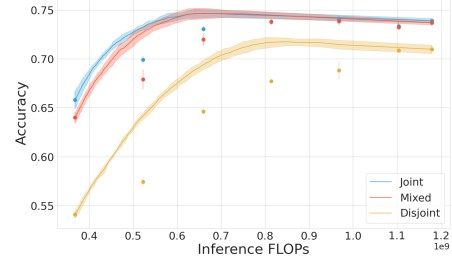

Figure 19: PBEE (36)

Figure 20: ZTW (29)

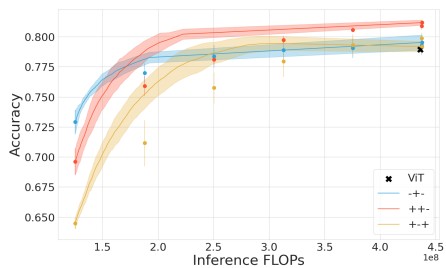
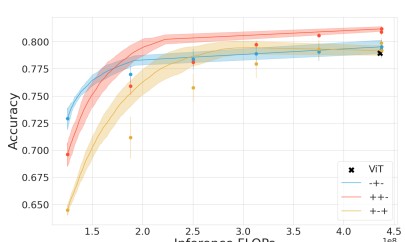

Figure 21: ViT / CIFAR-10 / GPF

Figure 22: ViT / CIFAR-10 / SDN

# D    REPRODUCIBILITY AND TRAINING DETAILS

For better reproducibility we include all the details of the experiments and analysis that we performed in this work. Upon publishing this work, we will also release the entire codebase.

## D.1    VIT

**Model set-up.**    The model's hyperparameters include a patch size of 4, a hidden dimension of 256, 7 layers, and 8 heads. We place internal classifiers after 2nd to 7th layers. The backbone achieves 79% accuracy.

**Training set-up.** During training, we use a batch size of 128 and the AdamW optimizer with no weight decay. For the backbone, we find learning rate of $5e-4$ to be optimal. For phases 2 and 3 we have performed a search through the following learning rates, $\{1e-3, 1e-4, 1e-5, 4e-3, 8e-3, 5e-4, 8e-4, 4e-5, 8e-5\}$.

## D.2 RESNET

**Training set-up.** In each training session, the SGD optimizer is utilized. For every training, the batch size is set to 125, and the hyperparameter learning rate is selected as the value from $\{8e-3, 2e-2, 5e-2, 8e-2, 2e-1\}$ which provide the best test classification performance.

**Model set-up.** ICs are placed in every other backbone block starting from the fifth block.

## D.3 EFFCIENTNET

**Training set-up.** We use SGD optimizer with cosine scheduler with early stopping with patience 10 epochs, momentum 0.9, batch size of 128 and gradient clipping at norm 1. We have performed a search through the following learning rates: $\{3e-1, 1e-1, 3e-2, 1e-2, 1e-3, 3e-3\}$ and we have found 0.1 to be the best.

## D.4 BERT

**Training set-up** We use AdamW optimizer with cosine scheduler for 10 epochs for Newsgroup dataset and 5 epochs for SST dataset, batch size of 32, weight decay 0.0001. We have performed a search through the following learning rates, $\{1e-6, 5e-6, 1e-5, 5e-5, 1e-4, 5e-4, 1e-3, 5e-3, 1e-2, 5e-2\}$

## D.5 NUMERICAL RANK

In this experiment, the ResNet-34 model is trained on the CIFAR-100 dataset using cross-entropy loss. The hyperparameters are taken from the run that provides the best performance when ResNet-34 is trained as described in subsection 5.1. The SGD optimizer is used with a learning rate of 8e-2, no learning rate scheduler, and a weight decay of 0.0. The Mixed and Joint regimes are trained using SDN under the same conditions, but with learning rates that provide the best performance within their respective regimes. All runs are trained with early stopping. Ranks are computed by creating a 2D matrix from tensors gathered right after the operations on a layer and before activation and batch normalization. The batch size dimension is kept as the first axis, and the subsequent dimensions are flattened into a single dimension, where the same 6000 features are randomly selected across the obtained matrices. The rank is then computed from the obtained matrices. The input used to obtain the tensors is the entire test set of the CIFAR-100 dataset. Similar results were observed on 10,000 randomly selected examples from the training set, chosen in a stratified manner.

## D.6 MUTUAL INFORMATION

In this experiment, the ResNet-34 model is trained on the CIFAR-100 dataset using cross-entropy loss. The hyperparameters are taken from the run that provides the best performance when ResNet-34 is trained as described in subsection 5.1. The SGD optimizer is used with a learning rate of 8e-2, no learning rate scheduler, and a weight decay of 0.0. The Mixed and Joint regimes are trained using SDN under the same conditions, but with learning rates that provide the best performance within their respective regimes. All runs are trained with early stopping. Mutual information is computed as described in the article [4]. The Jensen approximation was used, but similar results were observed with the Monte Carlo approximation. The input used to obtain the tensors is the entire test set of the CIFAR-100 dataset. Similar results were observed on 10,000 randomly selected examples from the training set, chosen in a stratified manner.

### D.7   LOSS LANDSCAPES

Experiments were performed for SDN architecture, with Resnet-20 as a backbone. The loss was calculated for the training dataset of CIFAR-10. For each loss landscape plot, the model is evaluated at 10,000 points in parameters space.

### D.8   MODE CONNECTIVITY

Experiments were performed for SDN architecture, with one model checkpoint chosen for each regime. Checkpoints for Mixed and Disjoint regimes use backbones from different seeds. Backbones are of Resnet-20 architecture with widen factor of 32 (1). The loss is calculated on the training dataset of CIFAR-10. In total, we visualize 22,500 points, each representing loss function for a model with weights lying in a plane defined by 3 points corresponding to 3 models (obtained by interpolation). One model is distinguished and is unchanged and other two are functionally equivalent (1) to the corresponding original models (obtained by permuting the weights with the weight matching algorithm)

