# OpenReview forum: "Joint or Disjoint: Mixing Training Regimes for Early-Exit Models"
_ICLR.cc/2025/Conference — Submitted to ICLR 2025_

### Official Review · Reviewer_nb8J · 2024-10-19

**Soundness:** 3
**Presentation:** 2
**Contribution:** 2
**Rating:** 5
**Confidence:** 5

**Summary:**

This paper analysis the

**Strengths:**

1. Early exiting is a very important research topic to achieve efficiency. Focusing this topic is valuable.

2. This paper provide a understanding of the early-exit neural networks, which may be useful for other researchers.

3. The author do both image and language experiment.

**Weaknesses:**

1. The novelty is limited. The proposed joint / disjoint / mixed training sound naive. Although the authors provide some analysis for early-exit networks, the proposed methods and experiments looks have few relation with these these analysis.

2. Inproper baseline network and evaluation choice. And I think it is the biggest problem. I am curious why the author mainly follow the practice in SDN [11], rather than follow the practice of MSDNet [9]. It is apparent that MSDNet have a more clean and reasonable architecture for early-exit, a very clean training setting, and a more systematic evaluation method for early-exiting.

    2.1)  The disadvantages of directly add early exits in resnet (as the practice in SDN) have been very througthly discussed in MSDNet paper. And MSDNet have a much stronger performance than SDN in a very clean training setting. I think the authors should do their experiments in more SoTA architectures.

    2.2) The line of MSDNet works [9, 7, 19, 32] provide a more reasonable evaluation method for early-exiting networks. They evaluation the networks in Budgeted Training and Dynamic Inference schemes. In the Budgeted Training scheme, they will calculate the threshold for each exits in the training set, and they use these thresholds to do evaluate in eval/test sets. However, the way SDN evaluate their model looks much naive. Furthermore, this submission "set 100 evenly spaced early-exit confidence thresholds" (as mentioned in line 319), is not very reasonable   compared with MSDNet.

3. The training setting is not clear and maybe infair. When the authors compare disjoint / joint / mixed training, it seems they have not keep the total training epoch (or some other method to evaluate training cost) the same. As a result, I am doubtful for their results.

4. The training hyper-parameter is also confuing. For example, in sec. D.3, the author claim they train 1500 epoches for efficientnet in line 791, while in line 791 they say they train efficientent for 200 epochs.

5. Lack of experiments.

    5.1) For image experiment, I think the results in imagenet-1k is very important. While the authors sometimes do experiments in CIFAR10, and sometimes in CIFAR100, limited ImageNet-1k results is provided.

    5.2) I also does not understand the way they choose CIFAR 10 or 100 in some small ablations.

    5.3) The authors do not compare they method with related works.


Minors:

1) Line 379: Imagenette --> ImageNet

2) Line 773: Imagenette --> ImageNet

**Questions:**

1. How the author claim " Disjoint and mixed regimes produce similar models, while the model trained in joint regime lies in a different basin" in Fig. 2? What is the x axis and y axis means in Fig. 2? If the distance in Fig. 2 mean something, it looks like the distance between the three points is similar. If you think a very high loss "mountain" separate  joint and the other two points, I think the loss "mountain" may means nothing in this space.

2. How the MODE CONNECTIVITY findings motivate the authors to design these methods?

3. How the numerical rank is computed in each layer? Why the rank will have a ~3000 rank? What network this experiment use?

4.  How the NUMERICAL RANK findings motivate the authors to design these methods?

I would raise my rating if the author give a reasonable explanation and necessary additional experiments for my comments and questions in the weakness section and questions section.

---

> ### Author Response · Authors · 2024-11-22
>
> We thank the reviewer for the time spent reviewing our work. Below, we present our responses to the issues raised. If our responses address the reviewer’s concerns, we would deeply appreciate consideration for raising the score. We are also available for further discussion at any time.
>
> > The novelty is limited. The proposed joint / disjoint / mixed training sound naive.
>
> We highlight that, until now, the multi-exit literature has been divided between works that train using the disjoint regime and those that train using the joint regime. This distinction was not emphasized in prior work, and it is often unclear from the text of a paper alone which regime was used.
> In our work, we introduce clear terminology to distinguish these approaches and demonstrate that the differences between joint and disjoint regimes can be substantial, even though the choice may initially appear straightforward or “naive.” Many works use the disjoint regime (e.g., [1–11]), perhaps without awareness that it leads to significantly reduced performance for smaller and medium computational budgets. We believe our comparison is a valuable and necessary contribution to the dynamic inference community. To the best of our knowledge, no prior work has focused on the impact of training regimes.
>
> > Improper baseline network and evaluation choice. And I think it is the biggest problem. I am curious why the author mainly follow the practice in SDN [11], rather than follow the practice of MSDNet [9]. It is apparent that MSDNet have a more clean and reasonable architecture for early-exit, a very clean training setting, and a more systematic evaluation method for early-exiting.
> > 2.1) The disadvantages of directly add early exits in resnet (as the practice in SDN) have been very througthly discussed in MSDNet paper. And MSDNet have a much stronger performance than SDN in a very clean training setting. I think the authors should do their experiments in more SoTA architectures.
>
> We thank the reviewer for highlighting this important point. Initially, we did not focus on architectures specifically designed for multi-exit models (MSDNet and RANet), as these are CNN-specific, whereas a significant portion of our experiments use transformer-based architectures. Nonetheless, we agree that including results on these architectures enhances the comprehensiveness of our study. Accordingly, we conducted analogous experiments comparing different regimes on the MSDNet architecture. The results, presented in Figure 6b of the revised manuscript, are consistent with our previous findings.
> As a minor note, we view SDN as an early exiting method that is independent of the base model architecture. For example, the SDN paper demonstrates its applicability across various CNN architectures, while [1] applies SDN to BERT. In contrast, MSDNet is inherently designed with multiple intermediate classifiers. While the MSDNet paper adopts a specific training and exiting strategy, alternative approaches, such as the patience-based method proposed in [1], could also be applied to the MSDNet architecture.
>
> > 2.2) The line of MSDNet works [9, 7, 19, 32] provide a more reasonable evaluation method for early-exiting networks. They evaluation the networks in Budgeted Training and Dynamic Inference schemes. In the Budgeted Training scheme, they will calculate the threshold for each exits in the training set, and they use these thresholds to do evaluate in eval/test sets. However, the way SDN evaluate their model looks much naive. Furthermore, this submission "set 100 evenly spaced early-exit confidence thresholds" (as mentioned in line 319), is not very reasonable compared with MSDNet.
>
> The evaluation method depends on the early exiting method used. In most of our experiments, we employ the SDN [13] early exiting method, which uses the confidence (max softmax probability) of an internal classifier and compares it to a threshold shared among all ICs. The GPF [14] method also uses a single threshold. By evaluating a broad range of confidence thresholds, we preserve the original evaluation method. For PBEE [12], we follow its original evaluation method, testing every possible integer patience threshold.
> In contrast, MSDNet [15] calculates individual per-IC confidence threshold values using a held-out development dataset. This approach is specific to the MSDNet line of works [15–19], which share similar early exiting and threshold-setting schemes.
> The literature on early-exit models is broad, and the scheme in [15] is just one example of an exiting strategy. Other methods include halting scores [20] or learnable controllers [21]. We already cover SDN, PBEE, GPF, and ZTW in our work, which we believe is sufficient given that our primary focus is on the training regimes rather than the exiting strategies.

---

> ### Author Response · Authors · 2024-11-22
>
> > The training setting is not clear and maybe unfair. When the authors compare disjoint / joint / mixed training, it seems they have not keep the total training epoch (or some other method to evaluate training cost) the same. As a result, I am doubtful for their results.
>
> We actually carefully designed the training procedure to ensure the fairness, and the best performance for each of the regimes. As the number of required epochs may vary between regimes due to different parameter groups the regimes optimize, we use early stopping with the patience hyperparameter set to a high value. We do report this both in the main paper and in the appendix. We also release the source code for full reproducibility.
>
> > The training hyper-parameter is also confusing. For example, in sec. D.3, the author claim they train 1500 epoches for efficientnet in line 791, while in line 791 they say they train efficientent for 200 epochs.
>
> Thank you for pointing this out. We mistakenly include the settings from the previous set-up. Now, as we mentioned earlier, we perform training with early exit for better convergence without explicitly setting the training length.
> > 5.1) For image experiment, I think the results in imagenet-1k is very important.
>
> In response to the suggestion, we have incorporated results on the ImageNet-1k dataset for the Vision Transformer architecture in Figure 7 of the revised manuscript. These results align with the previous findings for the mixed and joint regimes, while demonstrating that the performance gap in the disjoint regime diminishes at larger budgets. We appreciate the recommendation, as these additional results significantly strengthen our work.
> > While the authors sometimes do experiments in CIFAR10, and sometimes in CIFAR100,
> > I also does not understand the way they choose CIFAR 10 or 100 in some small ablations.
> Thank you for pointing out this issue to us. We have replaced the experiments on CIFAR-10 with new, analogous experiments on CIFAR-100 in the revised manuscript.
> > The authors do not compare they method with related works.
>
> We evaluate all three regimes across multiple architectures, datasets, and modalities. In Section 4.2, we test several early-exiting methods under the three regimes. Additionally, in the revised manuscript, we have included Section 4.5, where we examine loss and gradient scaling methods. If the reviewer still finds our evaluation insufficiently comprehensive, we would greatly appreciate more detailed guidance regarding which related works should be considered most relevant to our study.
>
> > Line 379: Imagenette --> ImageNet
>
>
> Imagenette is subset of the original ImageNet. In the revised manuscript we add descriptions of all the used datasets in the appendix.
>
> > How the author claim " Disjoint and mixed regimes produce similar models, while the model trained in joint regime lies in a different basin" in Fig. 2? What is the x axis and y axis means in Fig. 2? If the distance in Fig. 2 mean something, it looks like the distance between the three points is similar. If you think a very high loss "mountain" separate joint and the other two points, I think the loss "mountain" may means nothing in this space.
>
> This figure corresponds to Figure 1 in [22]. Figure represents interpolation between models obtained by training them using different regimes, represented by three red points in the Figure. Each point in the figure represents a model whose weights are a linear combination of the weights of the three considered models. Let red points have coordinates (x1, y1), (x1, y2), (x3, y3). Then there exist numbers a,b,c satisfying a+b+c=1 for which a point (x,y) in the figure satisfies (x,y) = a*(x1, y1) + b* (x1, y2) + c* (x3, y3). Then this point represents a model M, whose weights are equal to a*M1 + b*M2 + c*M3, where M1, M2, M3 are the weights of models represented by points (x1, y1), (x1, y2), (x3, y3), respectively. The loss ‘mountain’ is meaningful in this space because the models are permuted by weight matching algorithm as described in [22]. The ‘mountain’ is the barrier in the theory presented in [22].

---

> ### Author Response · Authors · 2024-11-22
>
> > How the MODE CONNECTIVITY findings motivate the authors to design these methods?
>
> Mode connectivity experiments show that the mixed regime arrives at a solution that lies in the same basin (as there is no ‘mountain’ between these two) as the solution to the disjoint regime. To us this is surprising, as the disjoint regime has significantly reduced performance for low and medium computational budgets, and thus we expected that the mixed regime and the joint regime would be more similar. However, as we show in Section 3.3 with the gradient dominance experiment, starting with a trained backbone effectively increases the impact of the last classifier to the point that the model never leaves the neighborhood of the minimum found during the first phase. This means that the mixed regime selects solutions that prioritize performance of the deepest classifiers, and this is consistent with the results in our experiments.
>
> > Although the authors provide some analysis for early-exit networks, the proposed methods and experiments looks have few relation with these these analysis.
> > …
> > How the NUMERICAL RANK findings motivate the authors to design these methods?
>
> We first demonstrate that the numerical rank in a backbone architecture (a standard neural network) starts with high-rank activations that progressively decrease toward the end of the network. Initially, we hypothesized that in early-exit architectures—composed of multiple sub-networks with separate exits—the rank reduction process would begin earlier in the network. Surprisingly, our study revealed the opposite: ranks in early-exit architectures tend to increase. We suspect this behavior is critical for the strong performance of early exits.
> This insight has implications for designing training regimes. A joint training regime is characterized by higher ranks at the beginning and lower ranks toward the end of the network, aligning with experimental results showing that joint training performs better under small computational budgets (corresponding to earlier exits). Conversely, in networks trained with a mixed regime, the numerical rank exhibits an opposite trend: lower in earlier layers and higher in later layers, leading to a flatter rank distribution (as described in the paper).
> As a result, mixed regimes consistently outperform joint regimes when higher computational budgets are available. These findings highlight the need for further research to explore the impact of early-exit training on intermediate representations in early-exit architectures.
> > How the numerical rank is computed in each layer?
>
> Ranks are computed by constructing a 2D matrix from tensors extracted immediately after layer operations and before applying activation or batch normalization. The batch size dimension is retained as the first axis, while the remaining dimensions are flattened into a single axis. From the resulting matrices, 6,000 features are randomly selected. The rank is then calculated based on these matrices. The input tensors are derived from the entire test set of the CIFAR-100 dataset (trained on ResNet-34). Similar results were observed using 10,000 stratified examples randomly selected from the training set.

---

> ### Author Response · Authors · 2024-11-22
>
> **References**
>
> [1] Panda, Priyadarshini, Abhronil Sengupta, and Kaushik Roy. "Conditional deep learning for energy-efficient and enhanced pattern recognition." 2016 design, automation & test in europe conference & exhibition (DATE). IEEE, 2016.
>
> [2] Bolukbasi, Tolga, et al. "Adaptive neural networks for efficient inference." International Conference on Machine Learning. PMLR, 2017.
>
> [3] Lahiany, Assaf, and Yehudit Aperstein. "Pteenet: post-trained early-exit neural networks augmentation for inference cost optimization." IEEE Access 10 (2022): 69680-69687.
>
> [4] Berestizshevsky, Konstantin, and Guy Even. "Dynamically sacrificing accuracy for reduced computation: Cascaded inference based on softmax confidence." International conference on artificial neural networks. Cham: Springer International Publishing, 2019.
>
> [5] Xin, Ji, et al. "DeeBERT: Dynamic Early Exiting for Accelerating BERT Inference." Proceedings of the 58th Annual Meeting of the Association for Computational Linguistics. 2020.
>
> [6] Leontiadis, Ilias, et al. "It's always personal: Using early exits for efficient on-device CNN personalisation." Proceedings of the 22nd International Workshop on Mobile Computing Systems and Applications. 2021.
>
> [7] Wołczyk, Maciej, et al. "Zero time waste: Recycling predictions in early exit neural networks." Advances in Neural Information Processing Systems 34 (2021): 2516-2528.
>
> [8] Li, Xiangjie, et al. "EENet: Energy Efficient Neural Networks with Run-time Power Management." 2023 60th ACM/IEEE Design Automation Conference (DAC). IEEE, 2023.
>
> [9] Xu, Guanyu, et al. "Lgvit: Dynamic early exiting for accelerating vision transformer." Proceedings of the 31st ACM International Conference on Multimedia. 2023.
>
> [10] Wang, Qingli, Weiwei Fang, and Neal N. Xiong. "TLEE: Temporal-wise and Layer-wise Early Exiting Network for Efficient Video Recognition on Edge Devices." IEEE Internet of Things Journal (2023).
>
> [11] Chataoui, Joud, and Mark Coates. "Jointly-Learned Exit and Inference for a Dynamic Neural Network." The Twelfth International Conference on Learning Representations. 2023.
>
> [12] Zhou, Wangchunshu, et al. "Bert loses patience: Fast and robust inference with early exit." Advances in Neural Information Processing Systems 33 (2020): 18330-18341.
>
> [13] Kaya, Yigitcan, Sanghyun Hong, and Tudor Dumitras. "Shallow-deep networks: Understanding and mitigating network overthinking." International conference on machine learning. PMLR, 2019.
>
> [14] Liao, Kaiyuan, et al. "A global past-future early exit method for accelerating inference of pre-trained language models." Proceedings of the 2021 conference of the north american chapter of the association for computational linguistics: Human language technologies. 2021.
>
> [15] Huang, Gao, et al. "Multi-Scale Dense Networks for Resource Efficient Image Classification." International Conference on Learning Representations. 2018.
>
> [16] Yang, Le, et al. "Resolution adaptive networks for efficient inference." Proceedings of the IEEE/CVF conference on computer vision and pattern recognition. 2020.
>
> [17] Han, Yizeng, et al. "Learning to weight samples for dynamic early-exiting networks." European conference on computer vision. Cham: Springer Nature Switzerland, 2022.
>
> [18] Yu, Haichao, et al. "Boosted dynamic neural networks." Proceedings of the AAAI conference on artificial intelligence. Vol. 37. No. 9. 2023.
>
> [19] Li, Hao, et al. "Improved techniques for training adaptive deep networks." Proceedings of the IEEE/CVF international conference on computer vision. 2019.
>
> [20] Figurnov, Michael, et al. "Spatially adaptive computation time for residual networks." Proceedings of the IEEE conference on computer vision and pattern recognition. 2017.
>
> [21] Dai, Xin, Xiangnan Kong, and Tian Guo. "EPNet: Learning to exit with flexible multi-branch network." Proceedings of the 29th ACM International Conference on Information & Knowledge Management. 2020.

---

> ### Author Response · Authors · 2024-12-01
>
> We thank the Reviewer for their detailed feedback. In this rebuttal, we have clarified our contributions, addressed the architecture and evaluation concerns, ensured fairness in training comparisons, and added new experiments, including on ImageNet-1K and MSDNet. We hope these updates address the Reviewer’s concerns.
>
> We would greatly appreciate it if the Reviewer could consider reevaluating their score based on our responses. If additional clarifications are needed, we are happy to provide them within the review timeline.

---

> ### Comment · Reviewer_nb8J · 2024-12-02
>
> Thank you to the authors for their rebuttal. The reviewer has the following further comments:
>
> 1. Disjoint training is evidently an inappropriate method for training early-exit networks, and Mixed training does not demonstrate significant benefits over Joint training. This is why I stated, "The proposed joint/disjoint/mixed training approaches seem naive." Moreover, L2W-DEN (reference [7] in this ICLR submission) and IMTA (reference [14] in this ICLR submission) have conducted novel research on training regimes and have explored Joint and Mixed training in meaningful ways. Specifically, L2W-DEN employs meta-learning to mimic the early-exit inference paradigm during training, while IMTA enhances collaboration between exits during training.
>
> 2. Experiments on MSDNet: I appreciate the additional experimental results provided for MSDNet.
>
> 3. Evaluation Methodology: The reviewer still believes that early-exit networks should be evaluated in the same manner as MSDNet. The reasons are twofold: (1) By calculating the threshold for each exit, the same model would achieve higher performance, as this is a more reasonable evaluation method. (2) This approach does not require retraining your model. You can simply use the open-sourced code from MSDNet and re-evaluate your model.
>
> 4. Training Hyperparameters: Comparing different methods with the same training epochs/resources is crucial for fair experimentation. While I agree that tuning different hyperparameters for different networks is another valid perspective, fair experiments cannot be overlooked. Additionally, if the authors use different training hyperparameters for different experiments, they should include the hyperparameter tuning results in the appendix to justify these choices and convince the reviewers.
>
> 5. The analysis provided in the submission still lacks important details. (I raised several questions in my previous review that the authors have answered, and there are apparently additional points that need clarification.) The relationship between these analyses and the proposed method should also be explained more thoroughly in future revisions or resubmissions.
>
> Overall, I have decided to retain my initial rating of "weak reject." I hope these comments will help the authors improve the quality of their revision or resubmission.

---

> ### Author Response · Authors · 2024-12-02
>
> We thank the reviewer for his comments. However, we are compelled to present our point of view. In particular, **we feel that the reviewer has not taken a deeper look at our rebuttal or the revised manuscript. Our impression is caused by the fact that in the last comment the reviewer refers to the [11], which is the very same work that we have added additional experiments for in our revised manuscript.** We detail our stance below.
>
> > Disjoint training is evidently an inappropriate method for training early-exit networks…
>
> We gently emphasize that this is **evident only in hindsight, and due to our experiments**. To the best of our knowledge, no other work has presented thorough experiments like ours to support this fact. To support our point of view, we highlight that a **significant fraction of multi-exit works uses the disjoint regime, e.g. [1, 2, 3, 4, 5, 6, 7, 8, 9, 10]**, which shows that this was not evident to the research community at all.
>
> > …Mixed training does not demonstrate significant benefits over Joint training.
>
> **This is simply not true.** In our rebuttal to other reviewers we have emphasized this aspect. On the ImageNet dataset the joint training results with significantly impaired results for medium and higher budgets (Figure 7 of the revised manuscript):
>
> | Regime   	| 25%             	| 50%             	| 75%             	| 100%            	| Max             	|
> |-----------|------------------|------------------|------------------|------------------|------------------|
> | Joint 	| **36.13**    	| 61.02        	| 67.24        	| 67.57        	| 67.57        	|
> | Mixed 	| 35.63        	| **62.07**    	| **70.50**    	| **70.91**    	| **70.91**    	|
>
> **The difference of almost 3 percentage points on ImageNet is definitely significant.**
>
> > Moreover, L2W-DEN (reference [7] in this ICLR submission) … conducted novel research on training regimes and have explored Joint and Mixed training in meaningful ways. … Specifically, L2W-DEN employs meta-learning to mimic the early-exit inference paradigm during training
>
> We point out that **the mechanisms proposed in the L2W-DEN paper [12] are completely orthogonal to the choice of the training regime** as defined in our work. L2W-DEN [12] trains the entire model jointly, but it might as well be used in the disjoint regime. On the other hand the authors of JEI-DNN [10], another work that focuses on the training-inference mismatch, freeze their backbone (disjoint), and achieve better results than L2W-DEN. Similarly, we are fairly sure that JEI-DNN could be used in the joint regime.
>
> While we acknowledge the importance of the training-inference mismatch in multi-exit works, it is definitely out of scope of our work. Evaluating every possible early-exit method in a single work is simply unfeasible, and we already provide a thorough empirical evaluation that tackles multiple aspects of mutli-exit models, e.g. IC placement, different methods, modalities, datasets, model types etc.
>
> > and IMTA (reference [14] in this ICLR submission) have conducted novel research on training regimes and have explored Joint and Mixed training in meaningful ways. … while IMTA enhances collaboration between exits during training.
>
> IMTA paper [11] proposes three approaches of enhancing training of multi-exit models: Gradient Equilibrium, Forward Knowledge Transfer, and Backward Knowledge Transfer. From these, the Gradient Equilibrium method can be considered somewhat similar to our proposed mixed regime. **In Section 4.5 of the revised manuscript we show that our proposed mixed regime achieves superior results over Gradient Equilibrium, and also eliminates the need to apply Gradient Equilibrium**.
>
> As for the other components, they are independent of the choice of the training regime as defined in our work. That is, e.g. distillation could be used both in the disjoint or joint regime. **We emphasize that both GPF [13] and ZTW [3] works propose similar mechanisms to those introduced in IMTA [11], and we do have experiments for both these methods.**

---

> > ### Author Response · Authors · 2024-12-02
> >
> > > Evaluation Methodology: The reviewer still believes that early-exit networks should be evaluated in the same manner as MSDNet. …
> >
> > We again emphasize the fact that there is nothing unique about the approach used by the MSDNet [14]. A different reviewer could raise a similar concern regarding e.g. lack of experiments for methods with a learned exit policy [15]. **We presented this argument in our previous answer, and yet the reviewer chose to ignore it and only restates his original statement.**
> >
> > > …By calculating the threshold for each exit, the same model would achieve higher performance
> >
> > Does the reviewer have any references to support this claim? A researcher should be sceptical about whether it would lead to a *significant* improvement without experimental evidence. Furthermore, we emphasize the fact that our work is not about setting or optimizing thresholds for early-exit works, and exploring whether one method of setting thresholds is superior to another is outside the scope of our work.
> >
> > > (2) This approach does not require retraining your model. You can simply use the open-sourced code from MSDNet and re-evaluate your model.
> >
> > This is not a valid **reason** for why early-exit networks “should be evaluated in the same manner as MSDNet”. It is a statement that eases evaluation at best. We kindly ask the reviewer to comment on why should this be **a reason** to use the method proposed in [14].
> >
> > A minor note – while we appreciate the reviewer’s hint about implementation, this statement is also not true. Tuning of the per-exit thresholds is performed on samples held-out from the training dataset. For an evaluation that is consistent with [14] we would have to retrain our models on the subset of the original train set.
> >
> > > Training Hyperparameters: Comparing different methods with the same training epochs/resources is crucial for fair experimentation. While I agree that tuning different hyperparameters for different networks is another valid perspective, fair experiments cannot be overlooked. Additionally, if the authors use different training hyperparameters for different experiments, they should include the hyperparameter tuning results in the appendix to justify these choices and convince the reviewers.
> >
> > We again emphasize that the hyperparameters are always the same for different regimes. We do not perform hyperparameter tuning - it would be computationally infeasible given the number of experiments we have.
> >
> > > The analysis provided in the submission still lacks important details. (I raised several questions in my previous review that the authors have answered, and there are apparently additional points that need clarification.) The relationship between these analyses and the proposed method should also be explained more thoroughly in future revisions or resubmissions.
> >
> > **We point out that this is an extremely vague answer**. We have invested significant amount of time and effort into answering the questions of the reviewer, and yet the reviewer writes:
> >
> > > still lacks important details
> >
> > without specifying which details are missing, and:
> >
> > > there are apparently additional points that need clarification
> >
> > without specifying which points need clarification. Lack of a detailed answer prevents us from addressing the reviewer’s further concerns and gives us zero value as feedback.

---

> > > ### Author Response · Authors · 2024-12-02
> > >
> > > ***References***
> > >
> > > [1] Berestizshevsky, Konstantin, and Guy Even. "Dynamically sacrificing accuracy for reduced computation: Cascaded inference based on softmax confidence." International conference on artificial neural networks. Cham: Springer International Publishing, 2019.
> > >
> > > [2] Xin, Ji, et al. "DeeBERT: Dynamic Early Exiting for Accelerating BERT Inference." Proceedings of the 58th Annual Meeting of the Association for Computational Linguistics. 2020.
> > >
> > > [3] Wójcik, Bartosz, et al. "Zero time waste in pre-trained early exit neural networks." Neural Networks 168 (2023): 580-601.
> > >
> > > [4] Lahiany, Assaf, and Yehudit Aperstein. "Pteenet: post-trained early-exit neural networks augmentation for inference cost optimization." IEEE Access 10 (2022): 69680-69687.
> > >
> > > [5] Panda, Priyadarshini, Abhronil Sengupta, and Kaushik Roy. "Energy-efficient and improved image recognition with conditional deep learning." ACM Journal on Emerging Technologies in Computing Systems (JETC) 13.3 (2017): 1-21.
> > >
> > > [6] Lattanzi, Emanuele, Chiara Contoli, and Valerio Freschi. "Do we need early exit networks in human activity recognition?." Engineering Applications of Artificial Intelligence 121 (2023): 106035.
> > >
> > > [7] Schuster, Tal, et al. "Confident adaptive language modeling." Advances in Neural Information Processing Systems 35 (2022): 17456-17472.
> > >
> > > [8] Li, Xiangjie, et al. "Predictive exit: Prediction of fine-grained early exits for computation-and energy-efficient inference." Proceedings of the AAAI Conference on Artificial Intelligence. Vol. 37. No. 7. 2023.
> > >
> > > [9] Xu, Guanyu, et al. "Lgvit: Dynamic early exiting for accelerating vision transformer." Proceedings of the 31st ACM International Conference on Multimedia. 2023.
> > >
> > > [10] Chataoui, Joud, and Mark Coates. "Jointly-Learned Exit and Inference for a Dynamic Neural Network." The Twelfth International Conference on Learning Representations. 2023.
> > >
> > > [11] Li, Hao, et al. "Improved techniques for training adaptive deep networks." Proceedings of the IEEE/CVF international conference on computer vision. 2019.
> > >
> > > [12] Han, Yizeng, et al. "Learning to weight samples for dynamic early-exiting networks." European conference on computer vision. Cham: Springer Nature Switzerland, 2022.
> > >
> > > [13] Liao, Kaiyuan, et al. "A global past-future early exit method for accelerating inference of pre-trained language models." Proceedings of the 2021 conference of the north american chapter of the association for computational linguistics: Human language technologies. 2021.
> > >
> > > [14] Huang, Gao, et al. "Multi-Scale Dense Networks for Resource Efficient Image Classification." International Conference on Learning Representations. 2018.
> > >
> > > [15] Dai, Xin, Xiangnan Kong, and Tian Guo. "EPNet: Learning to exit with flexible multi-branch network." Proceedings of the 29th ACM International Conference on Information & Knowledge Management. 2020.

---

### Official Review · Reviewer_k5RH · 2024-10-27

**Soundness:** 2
**Presentation:** 2
**Contribution:** 2
**Rating:** 5
**Confidence:** 4

**Summary:**

This paper studies different learning regimes (joint, disjoint) that could be followed when training the base model (backbone) and the additional internal classifiers. In this regard, the paper proposes a “mixed” regime, which follows a warming-up type of approach, where the backbone is first trained, and then, the internal classifiers are added and trained together with the backbone.

On the more theoretical side, the paper analyses the learning dynamics behind these regimes. On the empirical side, experiments on image and text classification problems based on several backbones show the capabilities of the proposed method at different computational budgets.

**Strengths:**

- A a high-level, the paper is very clear. There are no  barriers getting on the way towards understanding the problem addressed by the paper and its proposed solution.

- The empirical validation of the proposed method covers different data modalities, i.e. images and text. As a beneficial consequence, different datasets (CIFAR’10/100, ILSVRC12, Newsgroups and ) and models/architectures. This helps the reader get a good overview of the capabilities of the proposed method.

- Results seem to be reported over different runs, i.e. 4 according to Sec. 4 (l.323)

- The empirical evaluation is complemented with a more theoretical analysis of the effect of the considered training regimes.

**Weaknesses:**

- W1: weak positioning; Good part of the related work (l.430-441) is centered on discussing Early Exiting Networks without focusing on the training regime aspect, the core of the contribution put forward by the paper.

- W2: While the proposed mixed regime seems to outperform the classical joint approach under some circumstances, the technical novelty seems to be relatively reduced and some what comparable to existing techniques used to train multi-component networks. A comparison wrt. to these could help position the proposed method and stress further its novel aspects.

- W3: From the reported results, the proposed method seems to be less suitable for the setting of interest, i.e. the one with reduced computational budget. Moreover, the improvement of the proposed mixed regime over the classical joint strategy following other exit strategies, e.g. the entropy exit criterion (Sec. 4.2, Fig, 10) does not seem be that clear anymore.

- W4: Some observations made by the paper seem rather anecdotal. For instance, in Sec. 3.1 (l.130-146) some observations are made regarding the relative locations between loss values from models  trained following the considered regimes. Similarly, in several places (l.125, l.267-269, etc.) there are some statements regarding performance of input samples with different level of difficulty (e.g. easier vs. difficult to classify). It is unclear however, how prevalent/frequent these observations hold in the different problems/models/datasets that are considered. A supporting quantification of this aspect would be a proper companion to these statements.

- W5: The proposed method seems to be currently tested only in classification problems. Experiments on regression problems would provide further evidence on the applicability of the proposed method.

- W6: The content of the paper is too verbal at times, a more formal presentation of the considered training regimes would make more clear what are the different factors that are behind and influence one or the other. This would also help throw further light into how training would be affected by the selection of one or the other regime.

- W7: In its current form, the paper provides almost no details on the classification problems (and related datasets) that were considered on the empirical evaluation. This would not only be desirable for unfamiliar readers, but it would also serve as a point to verify whether the paper follows the standard or its own protocols, and ensure reproducibility of the reported results.

- W8: There are some inconsistencies in how models/datasets are used in some of the reported experiments. For instance, in some cases only specific model/dataset combinations are considered (Sec. 4.1 Fig.6 & 7). In other cases,  a given model. e.g. ViT is only trained on CIFAR-10 (Fig. 8) and in other cases on CIFAR-100 (Fig. 9). Results from Sec. 4.3 Fig.11 are limited only to the ViT model and CIFAR-10 dataset. A similar focus occurs on (Sec. 4.4, Table 1). Given this, it is hard to assess to what level the difference in performance are generalizable accross other settings that the specific combinations reported in the paper.

**Questions:**

[Suggestion] Regarding W1 and W2, a positioning wrt. to the iterative approaches like those used in GANs (Goodfellow, 2014) , R-CNN based detectors (Ren, 2017), and other multi-component models (Haidar, 2023) would be beneficial in this context?

[Suggestion] Regarding W4, quantifying how prevalent the stated observations are present in the conducted experiments could provide better grounds to support such statements. In a similar manner, I would suggest defining the difficulty of the samples of the considered datasets, quantify where these sample groups exit in the models and find the relationship of this wrt. the considered regimes.

[Suggestion] Regarding W8, I would suggest conducting experiments on all the possible combinations of the considered datasets/models. Certainly the page limitations will not allow adding all of them in the body of the paper, but the additional/supporting results could be part of the supplementary material.

References

- Goodfellow et al., "Generative Adversarial Nets",  NeurIPS 2016

- Ren et al., "Faster R-CNN: Towards Real-Time Object Detection with Region Proposal Networks", Transactions of Pattern Recognition and Machine Intelligence (T-PAMI) 2017

- Haidar et al., "Training Methods of Multi-label Prediction Classifiers for Hyperspectral Remote Sensing Images", Remote Sensing 2023.

**Details Of Ethics Concerns:**

N.A.

---

> ### Author Response · Authors · 2024-11-22
>
> We thank the Reviewer for the time spent reviewing our work. Below, we present responses to the issues listed by the Reviewer. If the Reviewer finds the answers satisfactory, we would greatly appreciate it if the Reviewer would consider raising the score. We also remain open to further discussion.
>
> > W1: weak positioning; Good part of the related work (l.430-441) is centered on discussing Early Exiting Networks without focusing on the training regime aspect, the core of the contribution put forward by the paper.
>
> Please note that we explicitly discuss the training regime aspect in lines 442-459. Nevertheless, as per the Reviewer’s suggestion, we have modified the related work section in our current revision to reduce the size of the paragraph about early-exit models in general (lines 430-441) and broaden the paragraphs about works using each regime. **In addition, we emphasize that existing works are almost completely oblivious to the training regime aspect, which is the main motivation behind our work.**
>
> > W2: While the proposed mixed regime seems to outperform the classical joint approach under some circumstances, the technical novelty seems to be relatively reduced and some what comparable to existing techniques used to train multi-component networks. A comparison wrt. to these could help position the proposed method and stress further its novel aspects.
> > ...
> > [Suggestion] Regarding W1 and W2, a positioning wrt. to the iterative approaches like those used in GANs (Goodfellow, 2014) , R-CNN based detectors (Ren, 2017), and other multi-component models (Haidar, 2023) would be beneficial in this context?
>
> We sincerely thank the reviewer for their insightful suggestion. The approach proposed in [1], which draws inspiration from GANs by alternating the training of two components over a predefined number of epochs, is indeed intriguing as an alternative training regime. However, the primary objective of our work is to investigate the impact of training regimes specifically employed for multi-exit models. To the best of our knowledge, existing early-exit works utilize either the joint or disjoint training regimes. While we recognize the potential of [1] as an interesting direction for future research, we believe this falls outside the scope of our current study.
>
> Instead, we believe that contextualizing the proposed mixed regime within existing approaches for training multi-exit models is a better way to strengthen our work. To this end, we have expanded our related work section and introduced Section 4.5 in the revised manuscript. In this section, we re-evaluate several existing methods that were proposed for enhancing **training** of multi-exit models. Specifically, **we reimplement two variants of loss scaling [2, 3] and gradient rescaling [4]**. These methods are directly relevant as they affect the training dynamics of multi-exit models, similarly to the change of the training regime. While these methods indeed result with performance improvements in the joint regime (for which they were originally developed), **they do not provide any gains for the mixed regime**.
>
> These findings underline the limitations of existing approaches and emphasize the novelty of our work, which provides a broader perspective on multi-exit models. We again thank the reviewer for their suggestion, as it significantly improved our work.
>
> Finally, we want to highlight that we see the technical simplicity of the proposed regime as an actual advantage rather than a disadvantage of our contribution. Straightforward, yet well-motivated methods that provide consistent improvements are crucial for advancing the field, and simplicity often facilitates broader adoption.

---

> ### Author Response · Authors · 2024-11-22
>
> > W3: From the reported results, the proposed method seems to be less suitable for the setting of interest, i.e. the one with reduced computational budget. Moreover, the improvement of the proposed mixed regime over the classical joint strategy following other exit strategies, e.g. the entropy exit criterion (Sec. 4.2, Fig, 10) does not seem be that clear anymore.
>
> We would like to clarify that we have never claimed the proposed mixed regime always improves performance across every possible budget. As demonstrated in Sections 3.1 and 3.3, the mixed regime indeed aims to enhance the performance of deeper ICs. Despite this, in some cases (e.g. Newsgroups or ResNet34 on CIFAR-100) the mixed regime provides superior performance in all budgets, while in the rest of our experiments it exhibits inferior performance only for the 10-20% lowest budgets.
>
> We respectfully disagree with the assertion that the lowest budgets are the most critical. The performance of multi-exit models typically deteriorates significantly at these levels across most tasks, and this reduces their practical relevance. However, if users wish to prioritize lower budgets, they can do so by weighting the losses of each IC, as demonstrated in Figure 15 of the revised manuscript.
>
> Finally, we emphasize that the mixed regime is not the sole contribution of our work. A key contribution lies in systematically analyzing the strengths and weaknesses of existing approaches in a principled and comprehensive manner. To the best of our knowledge, no prior work has thoroughly examined the impact of the choice of training regime.
>
> > W4: Supporting quantification of performance of input samples with different level of difficulty (e.g. easier vs. difficult to classify).
>
> As requested, we present a numerical analysis to quantify the 'hardness' of a dataset in the context of early-exit mechanisms. Specifically, we calculate the average FLOPs incurred during model inference when the data meets a specified confidence threshold. Comparing CIFAR-10 (Table 1a) and CIFAR-100 (Table 1b), our results indicate that, regardless of the threshold, CIFAR-100 samples consistently require more computation to achieve the same confidence level and exit, compared to CIFAR-10 samples. We define the 'hardness' of a dataset by the computational effort needed. Therefore, CIFAR-100 can be considered 'harder' on average than CIFAR-10. All experiments were conducted using the same architecture, ResNet-34, with hyperparameter optimization and model training performed to convergence to ensure a fair comparison. For completeness, we also provide the average accuracies for both datasets (Tables 2a and 2b).
>
>
> ### Table 1a: Cifar 10 - Flops x$10^8$ for chosen values of thresholds
>
> |     	| 0.2             	| 0.4             	| 0.6             	| 0.8             	|
> |---------|---------------------|---------------------|---------------------|---------------------|
> | Joint   | **364.46** ± 0.00	| **364.63** ± 0.11   | **368.09** ± 0.98   | **377.10** ± 3.25   |
> | Disjoint| **364.46** ± 0.00	| **368.67** ± 0.23   | **392.84** ± 0.79   | **442.74** ± 1.75   |
> | Mixed   | **364.46** ± 0.00	| **364.71** ± 0.09   | **368.51** ± 0.10   | **377.85** ± 0.42   |
>
> ### Table 1b: Cifar 100 - Flops x$10^8$ for chosen values of thresholds
>
> |     	| 0.2             	| 0.4             	| 0.6             	| 0.8             	|
> |---------|---------------------|---------------------|---------------------|---------------------|
> | Joint   | **364.74** ± 0.03	| **378.98** ± 0.70   | **419.54** ± 2.68   | **497.48** ± 4.93   |
> | Disjoint| **374.55** ± 0.72	| **448.47** ± 2.34   | **558.77** ± 3.61   | **699.39** ± 7.50   |
> | Mixed   | **364.62** ± 0.13	| **377.68** ± 1.10   | **417.00** ± 2.48   | **489.88** ± 5.05   |
>
> ### Table 2a: Cifar 10 - Accuracy for chosen values of thresholds
>
> |     	| 0.2             	| 0.4             	| 0.6             	| 0.8             	|
> |---------|---------------------|---------------------|---------------------|---------------------|
> | Joint   | **90.13** ± 1.52 	| **90.15** ± 1.50	| **90.54** ± 1.29	| **91.07** ± 1.06	|
> | Disjoint| **82.59** ± 0.32 	| **83.06** ± 0.37	| **85.89** ± 0.17	| **89.73** ± 0.37	|
> | Mixed   | **90.42** ± 0.42 	| **90.44** ± 0.43	| **90.85** ± 0.44	| **91.66** ± 0.30	|
>
> ### Table 2b: Cifar 100 - Accuracy for chosen values of thresholds
>
> |     	| 0.2             	| 0.4             	| 0.6             	| 0.8             	|
> |---------|---------------------|---------------------|---------------------|---------------------|
> | Joint   | **67.80** ± 0.44 	| **68.82** ± 0.32	| **71.24** ± 0.29	| **73.74** ± 0.32	|
> | Disjoint| **53.34** ± 0.21 	| **58.46** ± 0.23	| **64.60** ± 0.11	| **69.73** ± 0.44	|
> | Mixed   | **66.90** ± 0.50 	| **67.82** ± 0.36	| **70.26** ± 0.25	| **72.89** ± 0.44	|

---

> ### Author Response · Authors · 2024-11-22
>
> > W5: The proposed method seems to be currently tested only in classification problems. Experiments on regression problems would provide further evidence on the applicability of the proposed method.
>
> We follow the accepted practice established by published dynamic inference works. As shown in the table below, the vast majority of recent or influential early-exit papers were accepted to top conferences without any experiments on regression datasets. This indicates that experiments on multiple classification datasets are considered sufficient given thorough evaluation across different datasets, architectures and modalities. In the updated manuscript we have added results for MSDNet architecture, ImageNet dataset, and experiments that show the impact of three different methods for enhancing training of multi-exit models. Moreover, we evaluate all three regimes with multiple different early-exit methods (SDN, PBEE, ZTW, GPF), and perform IC placement and size analysis. As such, we believe that lack of regression experiments should not be considered as a significant weakness of our work.
>
> | Paper                                                                                        	| Experiments            	| Datasets                                       	| Architectures                  	| Year | Citations
> |--------------------------------------------------------------------------------------------------|----------------------------|----------------------------------------------------|------------------------------------|------|-----------
> | BranchyNet: Fast Inference via Early Exiting from Deep Neural Networks                       	| Classification         	| MNIST, CIFAR-10                                	| LeNet, AlexNet, ResNet         	| 2016 | 1273
> | Multi-Scale Dense Networks for Resource Efficient Image Classification                       	| Classification         	| CIFAR-100, ImageNet                            	| MSDNet, ResNet, DenseNet       	| 2018 | 871
> | Understanding and mitigating network overthinking (SDN)                                      	| Classification         	| CIFAR-10, CIFAR-100, TinyImageNet              	| VGG, ResNet, MobileNet, WideResNet | 2019 | 316
> | Improved Techniques for Training Adaptive Deep Networks                                      	| Classification         	| CIFAR-10, CIFAR-100, ImageNet                  	| WideResNet, ResNet, MobileNet  	| 2019 | 164
> | Distillation-Based Training for Multi-Exit Architectures                                     	| Classification         	| CIFAR-100, ImageNet                            	| MSDNet                         	| 2019 | 209
> | Bert loses patience: Fast and robust inference with early exit                               	| Classification, Regression | GLUE, SST-2, MNLI, STS-B                       	| ALBERT, BERT                   	| 2020 | 321
> | FastBERT: a Self-distilling BERT with Adaptive Inference Time                                	| Classification         	| Ag.News, Amz.F, DBpedia, Yahoo, Yelp.F, and Yelp.P | BERT                           	| 2020 | 365
> | Resolution Adaptive Networks for Efficient Inference                                         	| Classification         	| CIFAR-10, CIFAR-100, ImageNet                  	| RANet, MSDNet, DenseNet, ResNet	| 2020 | 265
> | Dual Dynamic Inference: Enabling More Efficient, Adaptive, and Controllable Deep Inference   	| Classification         	| CIFAR-10, CIFAR-100, ImageNet                  	| ResNet-50, MobileNet           	| 2020 | 88
> | Zero time waste: Recycling predictions in early exit neural networks                         	| Classification         	| CIFAR-10, CIFAR-100, ImageNet                  	| ResNet, WideResNet             	| 2021 | 43
> | A Global Past-Future Early Exit Method for Accelerating Inference of Pre-trained Language Models | Classification         	| GLUE datasets                                  	| BERT, ALBERT                   	| 2021 | 41
> | Learning to weight samples for dynamic early-exiting networks                                	| Classification         	| CIFAR-10, CIFAR-100, ImageNet                  	| MSDNet, RANet                  	| 2022 | 49
> | Boosted Dynamic Neural Networks                                                              	| Classification         	| CIFAR-10, CIFAR-100, ImageNet                  	| ResNet, VGG, WideResNet        	| 2023 | 13
> | Fixing Overconfidence in Dynamic Neural Networks                                             	| Classification         	| CIFAR-100, ImageNet, Caltech-256               	| MSDNet                         	| 2024 | 12
> | Jointly-Learned Exit and Inference for a Dynamic Neural Network                              	| Classification         	| CIFAR-10, CIFAR-100, ImageNet                  	| ResNet-50, MobileNet           	| 2024 | 2

---

> ### Author Response · Authors · 2024-11-22
>
> > W6: The content of the paper is too verbal at times, a more formal presentation of the considered training regimes would make more clear what are the different factors that are behind and influence one or the other. This would also help throw further light into how training would be affected by the selection of one or the other regime.
>
> We updated the manuscript with a more formal definition of the regimes. We separate these definitions from some practical takeaways where we describe how the choice of training regime may affect the training (they can be found in the Conclusion section at the end).  Please let us know if we could further improve this content or provide any other explanations.
>
> > W7. Here we provide a short description of each dataset used. We also include this information in the appendix of the updated manuscript.
>
> CIFAR-10: A dataset consisting of 60,000 color images of size 32x32 pixels, divided into 10 classes such as airplanes, cars, birds, and cats. It includes 50,000 training images and 10,000 test images, commonly used for benchmarking image classification algorithms.
> CIFAR-100: Similar to CIFAR-10 but with 100 classes containing 600 images each. Each image is a 32x32 pixel color image. The dataset is split into 500 training images and 100 testing images per class, providing a more challenging task due to the increased number of categories.
> ImageNet-1K: The dataset used in the ImageNet Large Scale Visual Recognition Challenge (ILSVRC), containing over 1.2 million training images across 1,000 classes. It serves as a standard benchmark for image classification and has spurred significant advancements in deep learning.
> TinyImageNet: A scaled-down version of the ImageNet dataset, containing 200 classes with 500 training images, 50 validation images, and 50 test images per class. Images are resized to 64x64 pixels, making it suitable for experimenting with deep learning models on limited computational resources.
> Imagenette: A subset of ImageNet consisting of 10 easily classified classes. Created to facilitate quick experimentation and benchmarking of image classification models without the computational overhead of the full ImageNet dataset.
> Stanford Sentiment Analysis: Refers to the Stanford Sentiment Treebank, a dataset of movie reviews with fine-grained sentiment labels. It includes 215,154 phrases in 11,855 sentences, allowing for detailed analysis of sentiment at both the phrase and sentence levels. SST-2 is a version of this dataset containing 2 classes.
> Newsgroup: The 20 Newsgroups dataset contains approximately 20,000 newsgroup documents evenly divided across 20 different topics. It is widely used for text classification and clustering tasks in natural language processing.
>
> > W8: There are some inconsistencies in how models/datasets are used in some of the reported experiments. For instance, in some cases only specific model/dataset combinations are considered (Sec. 4.1 Fig.6 & 7). In other cases,  a given model. e.g. ViT is only trained on CIFAR-10 (Fig. 8) and in other cases on CIFAR-100 (Fig. 9). Results from Sec. 4.3 Fig.11 are limited only to the ViT model and CIFAR-10 dataset. A similar focus occurs on (Sec. 4.4, Table 1). Given this, it is hard to assess to what level the difference in performance are generalizable accross other settings that the specific combinations reported in the paper.
> >...
> > [Suggestion] Regarding W8, I would suggest conducting experiments on all the possible combinations of the considered datasets/models. Certainly the page limitations will not allow adding all of them in the body of the paper, but the additional/supporting results could be part of the supplementary material.
>
> Thank you for bringing this issue to our attention. In this study, our goal was to conduct a broad range of experiments across various datasets and architectures to provide more comprehensive evidence. However, we acknowledge that this approach may make comparative analysis across the given settings more challenging. To ensure a more consistent presentation, we have revised the manuscript to use a common CIFAR-100 dataset baseline for the main experiments. Additional experiments covering a wider range of setups are now included in the Appendix.
>
>
> ***References***
>
> [1] Haidar, Salma, and José Oramas. "Training methods of multi-label prediction classifiers for hyperspectral remote sensing images." Remote Sensing 15.24 (2023): 5656.
>
> [2] Kaya, Yigitcan, Sanghyun Hong, and Tudor Dumitras. "Shallow-deep networks: Understanding and mitigating network overthinking." International conference on machine learning. PMLR, 2019.
>
> [3] Han, Yizeng, et al. "Learning to weight samples for dynamic early-exiting networks." European conference on computer vision. Cham: Springer Nature Switzerland, 2022.
>
> [4] Li, Hao, et al. "Improved techniques for training adaptive deep networks." Proceedings of the IEEE/CVF international conference on computer vision. 2019.

---

> ### Comment · Reviewer_k5RH · 2024-11-27
> **Re:Rebuttal**
>
> I value the attention given the to my review, and thank the authors for the efforts made to address my concerns
>
> **W1:** Thanks for taking action, however, the provided  extensions are reduced as to provide significant additional insight. Moreover, in the recently-published survey from [Rahmath et al., 2024], there is reference to a “two-stage” training strategy that resembles the proposed “merged” strategy (which would reduce the novelty put by this paper on that aspect). Moreover, it includes additional regimes (i.e strategies) not covered in the paper under review. While the work from [Rahmath et al., 2024] is very recent, the methods related to the “two-stage” additional training regimes.are not. It is unfortunate these regimes have not being included in the paper under review as it would have provided a complete analysis of the existing training regimes.
>
>
> **W2:** Thanks, the new experiments provide additional insights on the strengths of the considered training regimes.
> “we want to highlight that we see the technical simplicity of the proposed regime as an actual advantage rather than a disadvantage of our contribution. Straightforward, yet well-motivated methods that provide consistent improvements are crucial for advancing the field, and simplicity often facilitates broader adoption.“
> - Completely  agree, reason for which, no criticism has been put forward in my review on the technical simplicity of the proposed method
>
>
> **W3:** thanks for the clarification. I agree, high predictive performance at the lowest-budget regimes might not be practically attainable. On the other hand, having the highest performance at the lowest budget possible (not necessarily the lowest in the presented plots) is the main motivation behind early-exit methods, so it is a region of interest.
>
>
> **W4:** I appreciate “hardness” being defined as this further clarifies some of the statements I pointed to in my original review. The provided evidence is solid and supports the statements I had concerns abou
>
>
> **W5:** I appreciate the provided list as it shows the prevalence of the the settings considered on empirical evaluation presented on the paper. Having said that, limiting the evaluation to the standard setting will only constrain the occurrence of the observed trends to that setting. Addressing a novel setting would had positioned the paper farther apart from existing efforts and, consequently, strengthened the observations/contributions made by the paper.
>
>
> **W6:** Thanks.
>
>
> **W7:** Thanks, If possible I would advise including a condensed version of the provided description in the main body of the paper.
>
>
> **W8:** Thanks, having a common dataset consistently tested throughout the paper assists establishing links across experiments.
>
> To conclude, I appreciate the addition insights and clarifications that have been provided and the efforts invested in addressing the concerns I put forward on my review. Based on this grounds I have updated my initial score.
>
> **References**
> - Haseena Rahmath P, Vishal Srivastava, Kuldeep Chaurasia, Roberto G. Pacheco, and Rodrigo S. Couto. 2024. Early-Exit Deep Neural Network - A Comprehensive Survey. ACM Comput. Surv. 57, 3, Article 75 (March 2025), 37 pages. https://doi.org/10.1145/3698767

---

> > ### Author Response · Authors · 2024-12-01
> >
> > We are grateful for the reviewer’s engagement. Below we address the two remaining concerns of the reviewer. In particular, we show that the cited early-exit survey [1] is misleading and of poor quality. Most of multi-exit works have been incorrectly assigned to the “training strategies” by the authors of [1] – we discuss this aspect thoroughly. We provide results for the branch-wise training regime. Finally, we conduct the regression experiment with PBEE [2], and show that the findings are consistent with the ones from the classification task.
> >
> > > Moreover, in the recently-published survey from [Rahmath et al., 2024], there is reference to a “two-stage” training strategy that resembles the proposed “merged” strategy (which would reduce the novelty put by this paper on that aspect).
> >
> > Looking at the description in Section 5.4 of [1] and Figure 7 of that work, we find that **this training strategy is their name for the disjoint regime from our work**.
> >
> > > Moreover, it includes additional regimes (i.e strategies) not covered in the paper under review. While the work from [Rahmath et al., 2024] is very recent, the methods related to the “two-stage” additional training regimes.are not. It is unfortunate these regimes have not being included in the paper under review as it would have provided a complete analysis of the existing training regimes.
> >
> > As we explained above, our proposed “mixed” regime is novel. Furthermore, we analyzed Section 5 of [1] and **found multiple issues, including major and minor errors**. After a thorough analysis, we show that **almost all multi-exit works use either the joint or disjoint training regime**, despite multiple papers being listed in each row of Table 5 of [1].
> > - **Almost all of the “branch-wise” works listed in Table 5 are not multi-exit model works**. In these works this training approach is used to train a standard static model. The only work where “early exits” are used is **[3], where an early exit mechanism is introduced for the task of enhancing the quality of compressed images. However, upon further inspection of this work, it appears it trains the model in a manner that resembles the joint regime** - see Section 3.4 of that paper. Finally, note that this setup is significantly different from early exit models considered in our work. This is emphasized by the fact that this work does not cite even a single other early exit work. **To the best of our knowledge, no mutli-exit work uses the “branch-wise” regime.**
> > - For the supposed “Separate” strategy works listed in the Table 5: 1) **[4] uses either the joint or disjoint training regime** according to the description from this paper - see Section 4.2 of that paper. 2) **[5] also uses the joint training regime** according to the first sentence from Section 2.2 of that paper. The authors do use the term “separate training”, but it is for their proposed performance model, which predicts the performance of the final multi-exit model. 3) **[6] uses the disjoint training regime** according to Section 4.3 of that paper (see also the description of the “independent training” in Section 3.2 of that paper). 4) **Training method proposed in [7] is equivalent to joint training** - see the final paragraph of Section 2.2 of that paper. Moreover, the paper is not a multi-exit work, as the technique was proposed to enhance the final performance of static models. 5) **[8] is not a multi-exit model work** . It is more similar to cascade classifier works and is not even focused on deep neural networks. 6) **[9] uses joint training**, and the paper proposes to train the loss weight of each IC instead of fixing it.
> > - **Other statements in that paper also can be incorrect.** For example, [1] places that MSDNet paper [10] among the works that use the branch-wise training strategy (text of Section 5.2). In reality, it used the joint training regime.
> > - They state that the disjoint regime is useful when pretrained models are used. We point out that the use of pretrained models does not preclude the use of any particular regime, as evidenced by our experiments on pretrained ViT.

---

> > > ### Author Response · Authors · 2024-12-01
> > >
> > > - In our opinion **treating IC knowledge distillation as a training strategy (regime) as done in [1] is wrong**, as these are two separate design decisions of the multi-exit approach. For example, [11] explores the performance of different distillation approaches using the disjoint training regime, while [12] adds distillation loss to ICs that are trained jointly (joint regime).
> > > - Finally, for the papers listed in “Hybrid” group: 1) **[13] is not a multi-exit work**, and it simply scales the learning rate of every layer in a different manner, i.e. it is not similar to any of the listed strategies. 2) [14] trains multiple models **jointly** with reinforcement learning algorithms. They also use model cascades instead of multi-exit models. 3) [15] is listed because of distillation, which we consider as a separate aspect to training regimes, as we have explained above. 4) [16] **uses the disjoint regime** in its [code](https://github.com/pachecobeto95/distortion_robust_dnns_with_early_exit/blob/main/experiments/distorted_training_b_mobilenet_caltech.py#L288-L292). 5) [17] combines channel/layer skipping with early-exits. The “two-stage” training is used for the skipping component, and the *early-exits are trained jointly with the backbone*. 6) [18] combines joint and “two-stage” training strategies instead of joint and “separate” strategies as suggested in [1].
> > >
> > > We are surprised that the authors of [1] made that many mistakes, especially because some of the older papers (e.g. [7]) were properly discussed by the previous early-exit survey [19]. **This emphasizes the need for works such as ours.**
> > >
> > > As of now we are fairly confident that **all multi-exit works used either the joint or disjoint training regime**. Nevertheless, **to increase the strength of our work, we conduct an additional experiment where we implement the “branch-wise” strategy.** As we are not allowed to update the manuscript, below we present a table with the result for ViT on the CIFAR-100 dataset (cost of the model up to the first IC is larger than 25% of the cost of the original model, so there are no scores to report for "25%"):
> > >
> > > | Regime     	| 25%             	| 50%               	| 75%               	| 100%              	| Max              	|
> > > |-------------|------------------|--------------------|--------------------|--------------------|-------------------|
> > > | Joint   	| -            	| **62.38 +/- 1.92** | 67.52 +/- 1.85 	| 67.60 +/- 1.78 	| 67.60+/- 1.78 	|
> > > | Mixed   	| -            	| **61.06 +/- 1.34** | **68.42 +/- 0.49** | **68.64 +/- 0.39** | **68.64+/- 0.39** |
> > > | Disjoint	| -            	| 48.42 +/- 0.80 	| 62.55 +/- 1.30 	| 65.12 +/- 1.93 	| 65.12+/- 1.93 	|
> > > | Branch-Wise | -            	| **61.78 +/- 0.37** | 62.82 +/- 0.42 	| 62.79 +/- 0.43 	| 62.79+/- 0.43 	|
> > >
> > > The results show that the branch-wise approach gives inferior results when compared to the joint or mixed regimes for middle and higher computational budgets. For us these results are not surprising, and intuitively we expect the advantage of end-to-end approaches (joint, mixed) to increase for larger models. These results might also explain why none of the multi-exit works use this approach, and why the interest in layer-wise training of static models has waned.
> > >
> > > > I appreciate the provided list as it shows the prevalence of the the settings considered on empirical evaluation presented on the paper. Having said that, limiting the evaluation to the standard setting will only constrain the occurrence of the observed trends to that setting. Addressing a novel setting would had positioned the paper farther apart from existing efforts and, consequently, strengthened the observations/contributions made by the paper.
> > >
> > > **We agree with the reviewer that regression experiments would strengthen our work even further. Accordingly, we perform the regime comparison experiment for regression**.  The table below shows the results for the regression variant of PBEE [2] on the STS-B dataset:
> > >
> > > | Regime   	| 25%              	| 50%              	| 75%              	| 100%             	| Max              	|
> > > |-----------|-------------------|-------------------|-------------------|-------------------|-------------------|
> > > | Joint 	| **2.51 +/- 0.05** | 1.54 +/- 0.60 	| 0.54 +/- 0.01 	| 0.55 +/- 0.02 	| 0.55 +/- 0.02 	|
> > > | Mixed 	| **2.49 +/- 0.06** | **0.83 +/- 0.13** | **0.52 +/- 0.00** | **0.50 +/- 0.01** | **0.50 +/- 0.01** |
> > > | Disjoint  | 4.10 +/- 0.45 	| 2.72 +/- 1.25 	| 1.31 +/- 0.69 	| **0.51 +/- 0.01** | **0.51 +/- 0.01** |
> > >
> > > The reported values are MSE on the testset (lower is better). We see that the results are similar to those from classification tasks – disjoint has a significant performance gap for lower budgets, and the proposed mixed regime is slightly but noticeably better than the joint regime. We again thank the reviewer for his valuable suggestion as it allowed us to significantly improve our work.

---

> > > > ### Author Response · Authors · 2024-12-01
> > > >
> > > > ***References***
> > > >
> > > > [1] Rahmath P, Haseena, et al. "Early-Exit Deep Neural Network-A Comprehensive Survey." ACM Computing Surveys (2022).
> > > >
> > > > [2] Zhou, Wangchunshu, et al. "Bert loses patience: Fast and robust inference with early exit." Advances in Neural Information Processing Systems 33 (2020): 18330-18341.
> > > >
> > > > [3] Xing, Qunliang, et al. "Early exit or not: Resource-efficient blind quality enhancement for compressed images." European Conference on Computer Vision. Cham: Springer International Publishing, 2020.
> > > >
> > > > [4] Chiu, Ching-Hao, et al. "Fair Multi-Exit Framework for Facial Attribute Classification." arXiv preprint arXiv:2301.02989 (2023).
> > > >
> > > > [5] Ebrahimi, Maryam, et al. "Combining DNN partitioning and early exit." Proceedings of the 5th International Workshop on Edge Systems, Analytics and Networking. 2022.
> > > >
> > > > [6] Lattanzi, Emanuele, Chiara Contoli, and Valerio Freschi. "Do we need early exit networks in human activity recognition?." Engineering Applications of Artificial Intelligence 121 (2023): 106035.
> > > >
> > > > [7] Lee, Chen Yu, et al. "Deeply-supervised nets." Journal of Machine Learning Research 38 (2015): 562-570.
> > > >
> > > > [8] Venkataramani, Swagath, et al. "Scalable-effort classifiers for energy-efficient machine learning." Proceedings of the 52nd annual design automation conference. 2015.
> > > >
> > > > [9] Wang, Meiqi, et al. "Dynexit: A dynamic early-exit strategy for deep residual networks." 2019 IEEE International Workshop on Signal Processing Systems (SiPS). IEEE, 2019.
> > > >
> > > > [10] Huang, Gao, et al. "Multi-Scale Dense Networks for Resource Efficient Image Classification." International Conference on Learning Representations. 2018.
> > > >
> > > > [11] Wójcik, Bartosz, et al. "Zero time waste in pre-trained early exit neural networks." Neural Networks 168 (2023): 580-601.
> > > >
> > > > [12] Phuong, Mary, and Christoph H. Lampert. "Distillation-based training for multi-exit architectures." Proceedings of the IEEE/CVF international conference on computer vision. 2019.
> > > >
> > > > [13] Brock, Andrew, et al. "FreezeOut: Accelerate Training by Progressively Freezing Layers." NIPS 2017 Workshop on Optimization: 10th NIPS Workshop on Optimization for Machine Learning. 2017.
> > > >
> > > > [14] Guan, Jiaqi, et al. "Energy-efficient amortized inference with cascaded deep classifiers." Proceedings of the 27th International Joint Conference on Artificial Intelligence. 2018.
> > > >
> > > > [15] Ilhan, Fatih, et al. "Adaptive Deep Neural Network Inference Optimization with EENet." Proceedings of the IEEE/CVF Winter Conference on Applications of Computer Vision. 2024.
> > > >
> > > > [16] Pacheco, Roberto G., Fernanda DVR Oliveira, and Rodrigo S. Couto. "Early-exit deep neural networks for distorted images: Providing an efficient edge offloading." 2021 IEEE Global Communications Conference (GLOBECOM). IEEE, 2021.
> > > >
> > > > [17] Wang, Yue, et al. "Dual dynamic inference: Enabling more efficient, adaptive, and controllable deep inference." IEEE Journal of Selected Topics in Signal Processing 14.4 (2020): 623-633.
> > > >
> > > > [18] Xin, Ji, et al. "BERxiT: Early exiting for BERT with better fine-tuning and extension to regression." Proceedings of the 16th conference of the European chapter of the association for computational linguistics: Main Volume. 2021.
> > > >
> > > > [19] Scardapane, Simone, et al. "Why should we add early exits to neural networks?." Cognitive Computation 12.5 (2020): 954-966.

---

> > > ### Comment · Reviewer_k5RH · 2024-12-02
> > >
> > > I thank the authors for the efforts invested in providing this follow up response.
> > > I will take it into account when making my final recommendation

---

### Official Review · Reviewer_6pMM · 2024-11-05

**Soundness:** 3
**Presentation:** 2
**Contribution:** 2
**Rating:** 5
**Confidence:** 5

**Summary:**

This paper presents a new method to improve model efficiency by early exits. Previous methods in this line usually train the backbone and head classifiers at the same time (joint scheme), or separately (disjoint scheme). This work argues that they will impair the performance, so they propose to train the backbone first and then both the backbone and head exit networks together, sort of a method in between of the previous two paths. Experiments across various architectures and datasets show the effectiveness of the method.

**Strengths:**

1. The early exits methods for model efficiency is practical and interesting. And their method has been motivated by grounded observations.
2. The method is associated with a theoretical analysis from the lens of mode connectivity. Although I think the "theoretical" part can be more grounded and rigorous, the intent and attempt are valuable.

3. Empirical results suggest the method is effective against other counterparts.

**Weaknesses:**

1. One problem with the experiments is that the paper does not include evaluations on relatively large-scale datasets like ImageNet-1K. Many papers have noticed that the conclusions on CIFAR is hard to generalize to ImageNet-1K, so the results on ImageNet-1K are encouraged.

2. Methodologically, the paper method looks too simple technically and too intuitive. One sign that the paper lacks *real* technical contribution is that it has zero equations - only one, if any, is in page 4 without indexing. The paper claims to "conduct theoretical analysis". Sorry to say it is hard to see where the "theory" is rigorously defined or introduced. With this missing, the paper has 9 pages, 1 page shy of the max 10 pages, which is of course okay, but somehow tells us that the paper appears to be rushed out.

3. In most results, the performance advantage over mixed training is quite marginal. Ie, the results are not strong.

4. Some of the results look strange. Why in Fig 7(a) does the disjoint scheme perform unusually better than the others at large FLOPs?

Minior writing or presentation issues:
- ”joint” regime -> “joint”， ”disjoint” regime ->“disjoint”  -- many of the quotes are in wrong format.

**==== Post Rebutal ====**

I thank the authors' response. Unfortunately, the presented new results are not convincing to me.  I mentioned before the results are not strong. The authors "respectfully disagree with this statement" and argued "in most of our results the mixed regime provides statistically significant improvements", with Fig. 9 as support. Also, they have the new ImageNet-1K results in Fig. 7.

The problem with these results is: they are all reported by the authors; critical details are unclarified, and the performance is far below the standard ones.
- The original Tiny-Vit on ImageNet-1K without pertaining can reach over 78% top1 accuracy (see https://www.ecva.net/papers/eccv_2022/papers_ECCV/papers/136810068.pdf, Fig. 1), but in this paper, the authors only report ~70% (see Fig 7 of this paper). I wonder if the experiment was conducted following the standards.
- Similar problem, in Fig. 1 of this paper, the reported Tiny Vit only reached ~54% accuracy on CIFAR100, which is unusually low too. And no details about how the Tiny Vit is adapted for the CIFAR100 dataset.

Without these critical details, the claimed performance advantage is hard to verify. And nearly all the comparison baselines are from the authors instead of the existing papers. There is no clear evidence so far that these results are trustworthy. Given this issue, and the shallow technical novelty, I maintain my score at weak rej.

**Questions:**

NA

---

> ### Author Response · Authors · 2024-11-22
>
> We thank the Reviewer for the time spent reviewing our work. Below we present the responses to the issues listed by the reviewer. Should the Reviewer find our responses satisfactory, we would be sincerely grateful if the reviewer considered raising the score. We are also happy to engage in further discussion if needed.
>
> > One problem with the experiments is that the paper does not include evaluations on relatively large-scale datasets like ImageNet-1K. Many papers have noticed that the conclusions on CIFAR is hard to generalize to ImageNet-1K, so the results on ImageNet-1K are encouraged.
>
> In response to the suggestion, we have incorporated results on the ImageNet-1k dataset for the Vision Transformer architecture in Figure 7 of the revised manuscript. These results align with the previous findings for the mixed and joint regimes, while demonstrating that the performance gap in the disjoint regime diminishes at larger budgets. We appreciate the recommendation, as these additional results significantly strengthen our work.
>
> > Methodologically, the paper method looks too simple technically and too intuitive.
>
> We consider the simplicity of our method to be a strength rather than a limitation, as straightforward approaches often enable broader adoption. Many influential techniques, such as the GELU activation function [1], mixup augmentation [2], and dropout [3], share this characteristic. To further illustrate this, we argue that Figure 16 of the revised manuscript compares our proposed mixed regime with Gradient Equilibrium, a more technically complex method introduced in [4], an influential early-exiting framework that performs training following the joint regime. Despite the greater complexity of Gradient Equilibrium, it gives inferior results to the proposed mixed regime that does not require any loss or gradient scaling.
>
> We would also like to note that it is not immediately clear that the mixed regime should be the preferred option. Pretraining a backbone network with significantly more parameters than the ICs could potentially disrupt joint training and mutual performance between the backbone and ICs. Hence, we intended to perform a deeper analysis of training regimes through the lens of more advanced concepts such as mode connectivity, numerical rank, or mutual information to delve into the reasons why one regime could perform better than the other.
>
> > One sign that the paper lacks real technical contribution is that it has zero equations - only one, if any, is in page 4 without indexing. The paper claims to "conduct theoretical analysis". Sorry to say it is hard to see where the "theory" is rigorously defined or introduced.
>
> As suggested by the reviewer, we have updated the manuscript to include a more formal definition of the regimes. Additionally, we have incorporated a more mathematical and rigorous description of the deep learning concepts used to analyze the training dynamics of early-exit architectures. Please let us know if there are further improvements we can make or if additional explanations are needed.
>
>
> > the paper has 9 pages, 1 page shy of the max 10 pages, which is of course okay, but somehow tells us that the paper appears to be rushed out.
>
> We followed the ICLR 2025 call for papers guidelines that encourage authors to submit papers with 9 pages of main content. We attempted to ensure the work is well-written, but if the Reviewer finds some elements appearing rushed, then we kindly ask for other suggestions. Our revised manuscript makes use of the additional space provided by the tenth page. Please also note that we had additional content in the appendix.

---

> ### Author Response · Authors · 2024-11-22
>
> > In most results, the performance advantage over mixed training is quite marginal. Ie, the results are not strong.
>
> We respectfully disagree with this statement. We believe that the impression of weak results may be caused by the way we present them (as FLOPs vs. performance plots). We emphasize that in most of our results the mixed regime provides statistically significant improvements. For example, on the Newsgroups dataset (Figure 9 of the revised manuscript), the improvement is over 1.5 percentage points on average. If we were to generate a table similar to those used in the SDN paper, it would show a clear and significant improvement:
>
>
> | Regime   | 25%   	| 50%   	| 75%   	| 100%  	| Max   	|
> |----------|-----------|-----------|-----------|-----------|-----------|
> | Disjoint | 56.57 	| 64.23 	| 68.96 	| 71.03 	| 71.64 	|
> | Joint	| **66.31** | 71.93 	| 74.71 	| 75.70 	| 75.77 	|
> | Mixed	| 65.76 	| **72.07** | **75.31** | **76.48** | **76.65** |
>
> And similarly, for MSDNet on CIFAR-100 (Figure 6b of the revised manuscript):
> | Regime   	| 25%            	| 50%            	| 75%            	| 100%           	| Max            	|
> |--------------|--------------------|--------------------|--------------------|--------------------|--------------------|
> | SDN disjoint | 56.57 +/- 1.51 	| 64.23 +/- 0.81 	| 68.96 +/- 0.59 	| 71.03 +/- 0.88 	| 71.64 +/- 0.98 	|
> | SDN joint	| **66.31 +/- 0.24** | **71.93 +/- 0.37** | 74.71 +/- 0.17 	| 75.70 +/- 0.26 	| 75.77 +/- 0.19 	|
> | SDN mixed	| 65.76 +/- 0.73 	| **72.07 +/- 0.59** | **75.31 +/- 0.81** | **76.48 +/- 0.80** | **76.65 +/- 0.73** |
>
> Moreover, we would like to emphasize that the main contribution of this work is to  identify a new problem: how early-exit architectures should be trained. We define two existing training regimes and propose a new mixed regime. Our comparative analysis shows that the mixed regime is often the most robust, though there are scenarios where the joint or disjoint regimes may be preferable. Notably, our goal is not to assert the mixed regime's superiority but to provide a fair comparison of training regimes. The mixed regime's frequent advantages emerge as a result of this analysis.
>
>
> > Some of the results look strange. Why in Fig 7(a) does the disjoint scheme perform unusually better than the others at large FLOPs?
>
> In [6] the authors experience a similar problem. In particular, in Section 4.2 of [6] the authors discover that all tested early-exit methods perform similarly on each GLUE dataset. They hypothesize that the reason for this is the extremely low number of classes (binary classification) in those tasks, and this is the reason they perform additional experiments on Newsgroups (20 classes), with additional analysis of this aspect in section B6. As the disjoint regime performs as expected on the Newsgroups dataset, we also assume that the results on SST are specific because binary classification is a challenging setting for early-exit models.
>
>
>
>
>
>
>
>
>
>
> ***References***
>
> [1] Hendrycks, Dan, and Kevin Gimpel. "Gaussian error linear units (gelus)." arXiv preprint arXiv:1606.08415 (2016).
>
> [2] Zhang, Hongyi, et al. "mixup: Beyond empirical risk minimization." arXiv preprint arXiv:1710.09412 (2017).
>
> [3] Srivastava, Nitish, et al. "Dropout: a simple way to prevent neural networks from overfitting." The journal of machine learning research 15.1 (2014): 1929-1958.
>
> [4] Li, Hao, et al. "Improved techniques for training adaptive deep networks." Proceedings of the IEEE/CVF international conference on computer vision. 2019.
>
> [5] Kaya, Yigitcan, Sanghyun Hong, and Tudor Dumitras. "Shallow-deep networks: Understanding and mitigating network overthinking." International conference on machine learning. PMLR, 2019.
>
> [6] Wójcik, Bartosz, et al. "Zero time waste in pre-trained early exit neural networks." Neural Networks 168 (2023): 580-601.

---

> ### Author Response · Authors · 2024-12-01
>
> We thank the Reviewer again for their thoughtful feedback and the time dedicated to reviewing our work. In this rebuttal, we have addressed all the points raised in detail, including conducting new experiments on ImageNet-1K, providing formal definitions and clarifying performance improvements with updated presentations, and utilizing the additional page for elaboration. We hope that these responses address the Reviewer's concerns comprehensively.
>
> We would be sincerely grateful if the Reviewer could kindly reconsider their evaluation and potentially reassess the score based on the updated manuscript. Should there be any further questions or clarifications required within the constraints of the review timeline, we are happy to respond promptly.

---

> ### Public Comment · ~Bartosz_Wójcik1 · 2025-02-05
> **Clarification for the post-rebuttal response**
>
> We thank the reviewer for the post-discussion response, which we were not able to address due to it appearing after the discussion period. In this post we wish to clarify two concerns that were raised by the reviewer.
>
> > The original Tiny-Vit on ImageNet-1K without pertaining can reach over 78% top1 accuracy (see https://www.ecva.net/papers/eccv_2022/papers_ECCV/papers/136810068.pdf, Fig. 1), but in this paper, the authors only report ~70% (see Fig 7 of this paper).
>
> The reviewer is confusing the ViT-T architecture [1] with the TinyViT architecture [2]. TinyViT has a significantly different architecture, with staged, hierarchical design [2] similar to Swin Transformer [3]. We perform experiments with ViT-T as the backbone architecture, not TinyViT. The ViT-T results that we obtain are similar to those from the original paper [1].
>
> > Similar problem, in Fig. 1 of this paper, the reported Tiny Vit only reached ~54% accuracy on CIFAR100, which is unusually low too. And no details about how the Tiny Vit is adapted for the CIFAR100 dataset.
>
> Poor performance of ViTs on small datasets is a known phenomenon [4], and our results are similar to the ones from [4]. We provide architecture details for ViT-T on CIFAR-10 and CIFAR-100 in Appendix D.
>
> ***References***
>
> [1] Touvron, Hugo, et al. "Training data-efficient image transformers & distillation through attention." International conference on machine learning. PMLR, 2021.
>
> [2] Wu, Kan, et al. "Tinyvit: Fast pretraining distillation for small vision transformers." European conference on computer vision. Cham: Springer Nature Switzerland, 2022.
>
> [3] Liu, Ze, et al. "Swin transformer: Hierarchical vision transformer using shifted windows." Proceedings of the IEEE/CVF international conference on computer vision. 2021.
>
> [4] Zhu, Haoran, Boyuan Chen, and Carter Yang. "Understanding why ViT trains badly on small datasets: an intuitive perspective." arXiv preprint arXiv:2302.03751 (2023).

---

### Author Response · Authors · 2024-12-04
**Discussion summary**

We sincerely thank all the reviewers for their valuable feedback and engagement during the discussion. In the discussion period we have significantly improved our paper by:
- Adding ImageNet-1k experiments, addressing Reviewer 6pMM’s request, which further underscore the differences in model effectiveness across training regimes.
- Revising the related work section as suggested by Reviewer k5RH for better clarity, comprehensiveness and contextualization.
- Including a new section on IC loss scaling and gradient equilibrium experiments, demonstrating that the proposed mixed regime eliminates the need for the complex gradient equilibrium method.
- Discussing branch/layer-wise training approaches and providing experiments demonstrating their limitations compared to other training regimes.
- Presenting regression task results, as requested by Reviewer k5RH, which align with our findings from classification tasks.

Our results highlight the strengths of our proposed mixed regime, which is simple to implement and effective. The proposed approach alleviates the weaknesses of the joint regime, and eliminates the need to apply the technically involved gradient equilibrium method. We emphasize that our work is the first to present an exhausive discussion supported by thorough empirical experiments on the multi-exit training regimes. As we have shown in the discussion period, almost all multi-exit works apply either the joint or the disjoint regime without a detailed explanation or discussion of this aspect. We believe our work fills a critical gap in prior literature, which makes it a valuable contribution to the field.

---

### Meta-Review · Area_Chair_oJdV · 2024-12-13

**Metareview:**

This study improves the existing early-exist training strategy by proposing a mixed training regime where the backbone is trained first, followed by the training of the entire multi-exit network, so both limitations of joint training and disjoint training can be alleviated. The paper is well-motivated with a clear presentation. Extensive empirical analysis and evaluation are conducted to demonstrate the effectiveness of the proposed method. However, a major concern is the significance and scope of the experiments, which are mentioned by all the three reviewers. Besides, despite extensive supportive explanations provided, the method itself is technically too simple with limited novelty, as raised by the three reviewers. The AC looks through the paper and all the discussions, and agree with the reviewers.

**Additional Comments On Reviewer Discussion:**

All three reviewers have concerns regarding the technical novelty and the experiments. The authors provide additional experiments and explain the significance of their method during the rebuttal period. But the concerns remain for the three reviewers.

---

### Decision · Program_Chairs · 2025-01-22

Reject